# BRAIN INSIGHTS IMPROVE RNNS' ACCURACY AND ROBUSTNESS FOR HIERARCHICAL CONTROL OF CONTINUALLY LEARNED AUTONOMOUS MOTOR MOTIFS

## ABSTRACT

We study the problem of learning dynamics that can produce hierarchically organized continuous outputs consisting of the flexible chaining of re-usable motor motifs from which complex behavior is generated. Can a motif library be efficiently and extendably learned without interference between motifs, and can these motifs be chained in arbitrary orders without first learning the corresponding motif transitions during training? This requires (i) parameter updates while learning a new motif that do not interfere with the parameters used for the previously acquired ones; and (ii) successful motif generation when starting from the network states reached at the end of any of the other motifs, even if these states were not present during training (a case of out-of-distribution generalization). We meet the first requirement by designing recurrent neural networks (RNNs) with specific architectures that segregate motif-dependent parameters (as customary in continual learning works), and try a standard method to address the second by training with random initial states. We find that these standard RNNs are very unreliable during zero-shot transfer to motif chaining. We then use insights from the motor thalamocortical circuit, featuring a specific module that shapes motif transitions. We develop a method to constrain the RNNs to function similarly to the thalamocortical circuit during motif transitions, while preserving the large expressivity afforded by gradient-based training of non-analytically tractable RNNs. We then show that this thalamocortical inductive bias not only acts in synergy with gradient-descent RNN training to improve accuracy during in-training-distribution motif production, but also leads to zero-shot transfer to new motif chains with no performance cost. Besides proposing an efficient, robust and flexible RNN architecture, our results shed new light on the function of motor preparation in the brain.

## 1 INTRODUCTION AND RELATION TO OTHER WORKS

Animals have the remarkable ability to efficiently learn and compose elaborate continuous behaviors, which often relies on the flexible chaining of 'motifs' - reproducible bouts of behavior - in response to hierarchical commands (Zimnik & Churchland, 2021; Geddes et al., 2018; Merel et al., 2019b). The mechanisms behind this are however not fully understood, and engineering controllers for dynamical systems that could perform such complex and structured continuous behaviors in the real world has been a long-standing challenge of robotics (Brooks, 1986; Prescott et al., 1999; Merel et al., 2019b). Such tasks involve two different computational operations: first, at coarse temporal intervals, the flexible selection of discrete and abstract action commands by a 'high-level controller'; and, second, the transmission of each abstract command to a 'lower-level controller' that continuously produces a corresponding command - a 'motif' - for the motor effector.

Many lines of work have used artificial neural networks (ANNs) to solve the first - discrete - computational operation (notably using tools from deep reinforcement learning, e.g. (Merel et al., 2019a; Frans et al., 2018; Dennis et al., 2020)) and have leveraged the ability of ANNs to perform well when using a rich training set that includes many motif sequences (Frans et al., 2018; Dennis et al., 2020; OpenAI et al., 2021; Xu et al., 2020). However, to the best of our knowledge, in the context of the second computational operation, there is a gap in the literature about how to design ANN mechanisms that enable zero-shot transfer to performing new continuous motif sequences from a library of

independently learned motifs. Here we will focus on this latter task, which notably highlights a remarkable human skill, as we can - for instance - learn to pronounce a new word and then immediately include it in arbitrary sentences. While this biological relevance has led the neuroscience community to start investigating related questions, this literature has focused on a hand-designed 'bottom-up' approach. This approach studies how networks can function in spite of strong constraints introduced e.g. to mimic the brain activity patterns or connectivity, and/or to ensure analytical tractability (Kao et al., 2020; Logiaco et al., 2019; Sussillo et al., 2015; Zimnik & Churchland, 2021; Ijspeert et al., 2013; Kulvicius et al., 2012). Importantly, this approach cannot determine whether specific network features are more generally advantageous for solving a task - instead, with this bottom-up approach, these chosen features could reflect constraints that arose through random evolutionary idiosyncrasies, or could become irrelevant for networks that cannot be designed through analytical insights but that can still be successfully trained with modern machine learning algorithms. Overcoming these limitations has implications for engineering and can provide new neuroscientific insight.

To address this knowledge gap, here, we use Recurrent Neural Networks (RNNs) whose computational power is not arbitrarily constrained. An RNN is a type of ANN which is a generic dynamical system, and which therefore naturally fits the desired characteristics of motor outputs. Consequently, ANNs are indeed regularly used for tasks requiring the production of continuous outputs, including in the context of robotics (Wyffels & Schrauwen, 2009; Sussillo & Abbott, 2009; Tani, 2003; Liu et al., 2019; Merel et al., 2019a; Maheswaranathan et al., 2019). We will examine the ability of RNNs to *(i)* independently learn motor motifs in order to to build a continuously expandable motif library; and *(ii)* flexibly chain motifs in arbitrary orders (Fig. 1a and see Appendix A.1 for a formal definition). In order to better dissect the mechanisms by which RNNs can fail or succeed at this task, we focus on the purest form of motor control through dynamics: the production of trajectories without needing the external anchor of a time-dependent sensory input. We refer to this as autonomous control, which is especially relevant when sensory input is too unreliable (Yeo et al., 2016; Shenoy et al., 2013; Brembs, 2021), but also in other cases such as the above-mentioned speech production. We will show that while it is possible to engineer RNNs to independently, extendably and efficiently learn to produce single motifs in response to discrete input commands, these RNNs are limited in their generalization ability during improvisation of motif sequences. We will then use insights from the mammalian thalamocortical motor system - notably the presence of a motor preparation phase before each motif (Zimnik & Churchland, 2021; Nashef et al., 2021; Logiaco et al., 2019). We will show that weaving in these insights into performance-optimized RNNs leads to both improved motif production accuracy (through a positive synergy with single motif gradient-descent training) and excellent robustness during generalization to motif sequencing.

## 2 TASK AND ARCHITECTURE DESIGN

Here, we study the ability of RNNs to fulfill the first requirement of our task: the ability to acquire an extendable library of autonomous motifs (Fig. 1a, left column). However, before describing these analyses, we want to clarify how we chose the motifs that the RNNs have to learn. First, we chose motifs of long durations – on the order of a thousand timesteps – so that they strongly leverage the above-mentioned autonomous capabilities of RNNs. Second, because we are interested in assessing the relative expressive power of the RNNs we study, we have designed two different types of motor motifs so that they constitute two 'difficulty' levels for the RNNs. Following recent analytical approaches for studying RNN dynamics (Schuessler et al., 2020b;a; Logiaco et al., 2019), we characterize the difficulty of motifs as the number of certain basis functions - complex exponentials that act similarly as different frequencies of a Fourier transform - needed to approximate a motif well through linear combination. Therefore, we define a set of oscillatory motifs that are relatively easy to produce, and a set of more difficult 'step' motifs (Fig. 1d,f-g, see Appendix A.2 for the full list of motifs used in this paper). We train motifs using gradient descent – specifically, ADAM (Kingma & Ba, 2015)). Our objective function is the mean square error between desired and actual output.

We will now study how RNNs can meet the requirement to learn a new motif without 'catastrophic interference' with the memory of previously learned motifs - which relates to the literature on continual learning (Kirkpatrick et al., 2017; Parisi et al., 2019; Yoon et al., 2020; Farajtabar et al., 2019). This line of work emphasizes avoiding interference between gradient updates used to train a network on many sub-tasks, while promoting the re-use of neural resources across sub-tasks (so that the network makes efficient use of its parameters). We will use an 'architectural' approach to

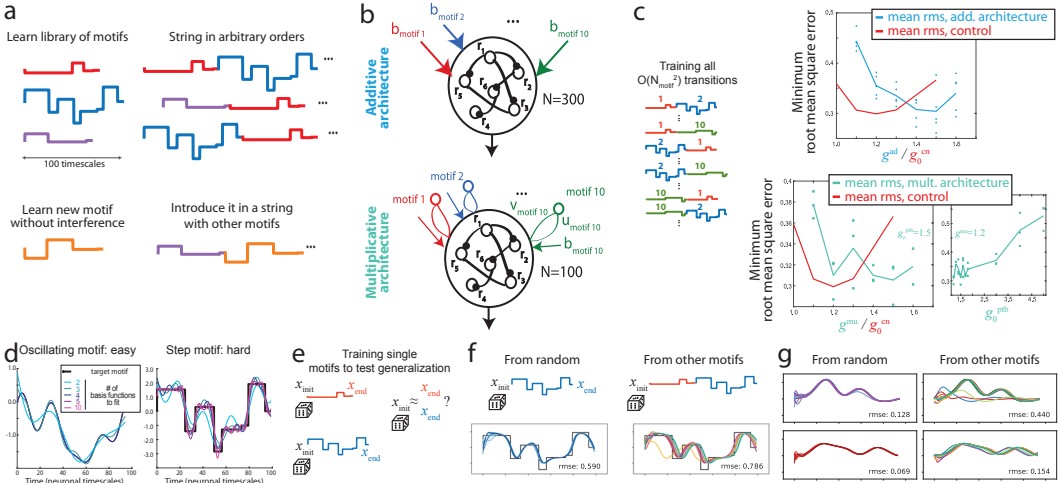

Figure 1: **Task and candidate networks. a**) Task. Left: learning an extendable library of motifs without interference. Right: without additional training, stringing the motifs into chains with arbitrary orders. **b**) Additive and multiplicative architectures, who may succeed in the task in **a** because they segregate parameters into motif-specific sets (schematized in colors) while benefiting from fixed shared recurrent and readout weights (schematized in black). **c**) Minimum root mean square error over training with overcomplete training set, depending on the gain hyperparameters $g^{ad}$, $g^{mu}$, and $g_o^{ptb}$. Dots are individual networks, the line is the average. In red, we show the mean minimum root mean square error in the control architecture (averaged over five individually trained networks for each $g_0^{cn}$). **d**) Examples easy oscillatory motif and hard step motif; fitting the latter requires a larger number of basis functions (complex exponentials) of varied oscillation frequencies. **e**) 'Classical' strategy to promote zero-shot transfer in order to produce a chain of motifs: train ANNs to produce the motifs starting from random initial network states $\mathbf{x}_{init}$ that emulate the variability of network states at the end of motifs $\mathbf{x}_{end}^{\mu}$ (where $\mu$ indexes the motif). **f**) Increased inaccuracy during zero-shot transfer to chains of motifs compared to when starting from random $\mathbf{x}_{init}$ drawn from the same distribution as during training (additive architecture). **g**) As in **f** but for oscillatory motifs. Note that the target trajectory (black) is buried below the output of the network (colored lines), because the latter is almost perfectly accurate with random $\mathbf{x}_{init}$.

continual learning that consists in segregating tuned parameters across the different motifs, because it will later facilitate the weaving of biological insights in our networks. With this approach, interference is fully prevented during sequential learning; but the proposed architectures may not be efficient, which is what we investigate below. To do so, we will compare these architectures to a 'standard' RNN with no segregation of tuned parameters. This RNN ('control architecture') *(i)* is fully-tuned, *(ii)* receives a static input $\mathbf{b}_\mu$ to instruct the motif $\mu$ (Maheswaranathan et al., 2019; Sussillo & Abbott, 2009; Tani, 2003), and *(iii)* is trained in a plausible noise-robust regime imposed by a random initialization of its state (see A.6.3). As expected given that this RNN has no protection against catastrophic interference (Kirkpatrick et al., 2017), we indeed find that when learning motifs sequentially, this RNN immediately forgets previously learned motifs when learning a new one (not shown) – even though it is possible to learn motifs by simultaneously including all of them in each training batch (Fig. 1c, and see below). This 'control architecture' obeys the following dynamics:

$$\tau \dot{\mathbf{x}} = -\mathbf{x} + g_0^{cn} \mathbf{J} \tanh(\mathbf{x}) + \mathbf{b}_\mu.$$

The output $y = \mathbf{w}^\intercal \tanh(\mathbf{x}) = \mathbf{w}^\intercal \mathbf{r}$ is produced through the vector $\mathbf{w}$ that is initialized from a centered Gaussian distribution with std $1/\sqrt{N^{cn}}$ – where here $N^{cn} = 50$ is the number of recurrent units of this control network, so that its outputs have appropriate maximal positive or negative magnitude scaling as $\sqrt{N^{cn}}$. Also, the recurrent interactions weights $\mathbf{J}$ are initialized with iid elements taken from a standard Gaussian as previous work has shown that when choosing a gain $g_0^{cn} > 1$ this leads to rich dynamical regimes appropriate for complex computations (Sompolinsky et al., 1988; Sussillo & Abbott, 2009; Sussillo et al., 2015; Schuessler et al., 2020b). We performed hyperparameter tuning on $g_0$. The dynamics are discretized using Euler's integration method where $dt = 0.1\tau$.

Table 1: Number of neurons and number of parameters in the different architectures

|  | **Additive** | **Multiplicative** | **Control** |
| --- | --- | --- | --- |
| # of recurrent units $N$ | 300 | 100 | 50 |
| # of learned parameters for 10 motifs | 3000 | 3000 | 3050 |
| # of motif-specific parameters per motif | 300 | 300 (input: 100, loop: 200) | 50 |

We now consider modifications of this control RNN that address this issue of catastrophic forgetting by segregating the parameters involved in learning different motifs while sharing common computational resources across motifs (Parisi et al., 2019; Merel et al., 2019a;b). Here, we will share *(i)* fixed readout weights (with the above-mentioned centered Gaussian distribution with std $1/\sqrt{N}$ as this is sufficient to ensure the successful production of our motifs); and *(ii)* the recurrent weights as they can set rich 'baseline dynamics' that can be modulated by some motif-specific weights. We globally adjust the shared recurrent weights through tuning their above-mentioned gain hyperparameter $g$ (a similar alternative could be to pre-tune these recurrent weights to an original set of motifs and to then freeze them, but - as we will see - our chosen approach leads to good results).

First, we consider an 'additive' architecture (Fig. 1b, top). Here, each motif $\mu$ is produced in response to learning the input vector $\mathbf{b}_\mu$, leading to the following dynamics for the network activities:

$$\tau \, \dot{\mathbf{x}} = -\mathbf{x} + g^{\text{ad}} \, \mathbf{J} \tanh(\mathbf{x}) + \mathbf{b}_\mu,$$

where the input acts as a motif-specific controller of the dynamics. Note that this occurs because the gradient of the loss with respect to the input weights propagates through the recurrent connections - whereas if different outputs weights would be learned for different motifs (Jaeger, 2007), the dynamics would not be affected by learning.

Second, we consider a 'multiplicative' architecture (Fig. 1b, bottom), that is inspired by both previous machine learning literature (Sutskever et al., 2011; Schuessler et al., 2020b) and the anatomy of the brain's motor system (Guo et al., 2017; Logiaco et al., 2019). Here, each motif $\mu$ is produced in response to both a learned input vector $\mathbf{b}_\mu$ and a learned rank-one perturbation of the connectivity $\mathbf{u}_\mu \mathbf{v}_\mu^\mathsf{T}$. The latter is equivalent to a loop through an instantaneous 'unit' receiving input from the recurrent network through the weights $\mathbf{v}_\mu$ and feeding back through the weights $\mathbf{u}_\mu$. Here, this motif-specific loop participates to modulating the dynamics (Logiaco et al., 2019), in a way that yields more computational flexibility compared to networks with random feedback weights (Susman et al., 2021). Interestingly, learning the full recurrent weights in randomly initialized RNNs can yield a low-rank weight update (Schuessler et al., 2020b). Therefore, by imposing that the motif-specific learning is restricted to a low-rank weight perturbation, we expect to get close to full-weight learning while enabling segregation of the learned weights per motif. The dynamics are:

$$\tau \, \dot{\mathbf{x}} = -\mathbf{x} + \left( g^{\text{mu}} \, \mathbf{J} + \mathbf{u}_\mu \mathbf{v}_\mu^\mathsf{T} \right) \tanh(\mathbf{x}) + \mathbf{b}_\mu,$$

where $g^{\text{mu}}$ and $\mathbf{J}$ are defined as for the additive network, and $\mathbf{u}_\mu$ and $\mathbf{v}_\mu$ are each learned and initialized iid from a centered Gaussian with std $g_o^{\text{ptb}}/\sqrt{N}$ (i.e. expected norm $g_o^{\text{ptb}}$).

To test the baseline relative accuracies of the additive, multiplicative and control networks, we trained them on all the possible chains of motifs of length two (excluding repetitions of the same motif) for the ten step motifs (Fig. 1c left). Consistent with the continual learning aspect of our task – which values limiting the number of parameters that need to be tuned and stored per motif – we equalized the number of tunable parameters across architectures (Collins et al., 2017). Interestingly, after optimizing over hyperparameters (i.e., $g^{\text{ad}}$, $g^{\text{mu}}$, $g_o^{\text{ptb}}$, and $g_0^{\text{cn}}$), we found that all networks had similar accuracy (Fig. 1c, reminiscent of (Collins et al., 2017)). This suggests that our strategy of segregating tunable parameters in the additive and multiplicative networks do not lead to drastic decrease in per-tuned-parameter expressivity, while by construction preventing interference when learning motifs sequentially (while the control network suffers from forgetting of previously learned motifs when learning new ones as expected, not shown). On the other hand, we note that the multiplicative architecture, which is closer to models constrained to mimic brain dynamics and architecture (Logiaco et al., 2019), appears to have similar accuracy as the additive network while requiring fewer neurons. This echoes recent results suggesting that more biologically-plausible

object-recognition ANNs tend to have architectures that require fewer neurons (Nayebi et al., 2021). We now turn to investigate the robustness of these different architectures when improvising motif sequences after being trained on single motifs – a training strategy which, by construction, enables learning new motifs without interference for the additive and multiplicative architectures.

## 3 BRITTLENESS OF STANDARD RNNS DURING GENERALIZATION

Here, we ask whether RNNs trained on single motifs can produce arbitrary chains of motifs – a form of zero-shot transfer where the RNN's internal state differs between training and testing due to the RNN's memory over several timesteps (a similar network mechanism as in (Lake & Baroni, 2017)).

We evaluate a standard technique used to promote both generalization and noise-robustness in ANNs, that relies on leveraging randomness (e.g. (Vezhnevets et al., 2017; Liu & Hodgins, 2017; Merel et al., 2019a)) during training. Each motif is trained in isolation but with initial network activities selected randomly according to a distribution that approximates the activities at the ends of all other motifs (Fig. 1e). In this scenario, if we can approximate the end-of-motifs distribution well enough and if training succeeds, then all transitions should work with no transition-specific training. Unfortunately, the end-of-motifs distribution is unknown, is training-dependent, and could be heterogeneous across motifs. However, since we are using networks with large $N$, Gaussian weights, and a $\tanh$ nonlinearity, it is known that an uncorrelated Gaussian vector can in some cases well-approximate this unknown distribution (Landau & Sompolinsky, 2018). Indeed, we observed that a standard normal is a good choice for the marginal statistics (Fig. 2a and b, panels (ii)). Hence, we tried training the RNNs to generate the step motifs one at a time when starting from random initial network activities $\mathbf{x}_{\text{init}}$ drawn from a standard normal distribution. We set our hyperparameters to the optima from Fig. 1c ($g^{\text{ad}} = 1.4$, $g^{\text{mu}} = 1.4$ and $g_o^{\text{ptb}} = 1.5$) and otherwise trained as above. For all architectures, training was successful (Fig. 2a and b panel (i), left; Fig. 2c left).

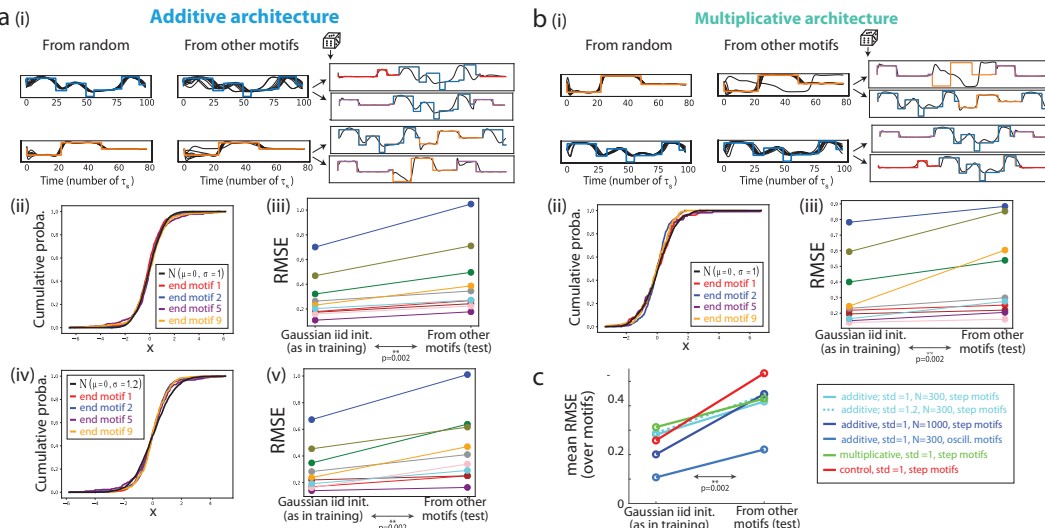

Figure 2: **Motif sequencing by RNNs trained on individual motifs from random initializations**. Step motifs are always used except in panel **c**. **a**) Additive architecture; **b**) multiplicative architecture. Panels (i): Network output (black traces, compared to the desired motif shown in color) when initial conditions $\mathbf{x}_{\text{init}}$ are randomly drawn from a standard Gaussian as during training vs. when transitioning from other motifs (with example full motif chains shown on the right). Panels (ii): After training with $\mathbf{x}_{\text{init}}$ drawn from a standard Gaussian, 'marginal' cumulative distribution of network states $\mathbf{x}_{\text{end}}^{\mu}$ at the end of example motifs. Panels (iii): for each motif, root mean square error taken over 9 random initializations (left) vs. when transitioning from 9 other motifs. P-values from two-sided Wilcoxon signed-rank test. Panels (iv) and (v): same as (ii) and (iii) but when training with $\mathbf{x}_{\text{init}}$ drawn from a Gaussian with standard deviation $1.2$. **c**) Average root mean square error over motifs during training conditions vs. during zero-shot transfer, for different architectures, network sizes, motif types, and variance of the random initialization. All comparisons are significant.

We then asked our networks to chain two motifs (the first one starting from random initial activities as in training), and found that the performance of the second motif was substantially degraded compared to the same motif produced when starting from random (Fig. 2a,b panels (i) and (iii); see also Fig. 1g right). Each transition from a given first motif leads to a relatively reproducible output for the second motif because the network variability is low at the end of the first motif. However the second motif's output can vary substantially from the target when following certain first motifs. Moreover, the inaccuracies are not restricted to the start of the second motif; rather, the whole output is either clearly shifted or even more grossly wrong. Note that these issues are also present in the control architectures and are therefore not caused by our strategy of segregating the tuned parameters per motif to tackle catastrophic interference (Fig. 2c, Appendix A.3.5).

Interestingly, the shape of the marginal distribution of the elements of $\mathbf{x}$ at the end of a given motif $\mu$ ($\mathbf{x}_{\text{end}}^{\mu}$) was not a good predictor of whether transitions from this motif would lead to worse performance. For instance, in the case of the additive network, the transition from motif 1 to motif 2 leads to poor performance for motif 2 (Fig. 2a (i), top right), even though the marginal distribution of $\mathbf{x}_{\text{end}}^{1}$ at the end of motif 1 appears extremely similar to a standard Gaussian (Fig. 2a (ii), red curve). Conversely the distribution of $\mathbf{x}_{\text{end}}^{5}$ at the end of motif 5 is wider than the standard Gaussian (Fig. 2a (ii), purple curve), but the transition from motif 5 to motif 2 leads to a more accurate output (Fig. 2a (i), second chain from top right). This strongly suggests that between-unit correlations impact transition success, which complicates sampling the various $\mathbf{x}_{\text{end}}^{\mu}$ when initializing motifs during training.

Figs. 2a,b panels (ii) show that the standard deviation of the marginal distribution of the values of $\mathbf{x}_{\text{end}}^{\mu}$ slightly exceeds 1 for some motifs. To ensure that the transition failures we observed when initializing with the standard normal were not due to this mismatch in scale, we retrained our additive network with initial $\mathbf{x}$ values sampled from a Gaussian with std of 1.2 (Fig. 2a panels (iv) and (v)). We observe that the end-of-motif distributions still have standard deviations around 1 for all motifs (Fig. 2a panel (iv)). Despite training with a wider distribution, performance impairment after transitioning still occurred with even slightly greater errors compared to training with a standard normal (Fig. 2a panel (iv) Fig. 2c dashed cyan line).

In addition, we verified that the decreased performance during zero-shot transfer was not a consequence of using our relatively difficult step motifs for which accuracy is limited even when starting from random. Using our oscillatory motifs instead (as in Fig. 1g), we see that performance when starting from random improves significantly, but zero-shot transfer is again unsuccessful with relatively large second motif errors during motif chaining (the light blue line in Fig. 2c, notice that the increase of error during zero-shot transfer is of similar amount as for the step motifs).

Finally, we also investigated whether larger networks would somehow be more regularized in a way that would favor zero-shot transfer performance. However, increasing the network size from $N = 300$ to $N = 1000$ did not improve the performance during zero-shot transfer; instead, as expected, it only improved accuracy when the network was initialized from random as in training (Fig. 2c, cyan line vs. dark blue line).

The failure to robustly transition between motifs when training on single motifs that are randomly initialized is a particular instance of an out-of-distribution generalization limitation, a general feature of ANNs (Russin et al., 2019; D'Amour et al., 2020). Not only is the shape of the true marginal distribution of the values of $\mathbf{x}_{\text{end}}^{\mu}$ at the end of a given motif not exactly matched to the statistics of the Gaussian distribution used during training, but we also neglect the correlations and other higher order interactions between the values of $\mathbf{x}_{\text{end}}^{\mu}$. As these complex $\mathbf{x}_{\text{end}}^{\mu}$ statistics are shaped during training, there is a self-consistency issue between initialization during training and resulting $\mathbf{x}_{\text{end}}^{\mu}$.

Note that trying to avoid this issue by including all transitions between motifs in the training set (Fig. 1c, left) would scale quadratically with the number of motifs and is thus ultimately prohibitive. Further, even including a few motifs transitions in the training set would disrupt the sequential learning strategy that prevents the training set to grow over time in a continual learning setting. Indeed, on the one hand, sequential training would require motif parameters to only be trained for transitioning from previously learned motifs - which would compromise the robustness of the motifs that were learned earlier, especially if the motifs' properties change over the 'life' of the agent. On the other hand, re-training earlier-learned motifs to improve their accuracy when they are initialized from later-learned motifs causes the training set to grow with the number of learned motifs, and runs the risk of modifying the network's end-of-motif activity for these re-trained motifs, which could in turn

affect the transitioning from these motifs to any others. Another naive attempt to solve the transition issue would be to train all motifs from a fixed arbitrary network state, and to implement a hard reset to this state before each motif. However, this strategy would introduce detours at transitions in case of discrepancy between two motif's end and start points – and would more generally prevent the RNN's dynamics to use the implicit information about the previous motif's shape encoded into its internal state in order to implement a smooth transition to the next motif. As a side note, training single motifs initialized with a reset from a noiseless fixed state would not be noise robust (Fig. 16a), while using a noisy reset actually corresponds to testing and training our networks on random initial conditions – and we will show below that, besides introducing detours at transitions, this naive strategy leads to impaired learning and poorer performance compared to our biologically-inspired solution. To address these issues, we will now leverage insights from the brain's motor network. This is promising because human behavior demonstrates that the brain has acquired inductive biases that enable zero-shot motor sequence improvisation with agile transitions between motifs.

## 4 BRAIN-INSPIRED INDUCTIVE BIASES IMPROVE ACCURACY & ROBUSTNESS

Recordings from the brain's motor cortex during flexible sequencing indicate that a period of motor preparation precedes every motif in a sequence (Zimnik & Churchland, 2021). This preparation leads the activity to quickly converge towards a motif-specific pattern (Lara et al., 2018). This motor preparation process involves interactions between the recurrent motor cortex and thalamus (Nashef et al., 2021) – a brain region whose neurons quickly respond to cortical input and then projects back to cortex, forming feedback loops that effectively act as perturbations of the effective cortical connectivity (Kao et al., 2020; Logiaco et al., 2019). In a computationally limited linear regime, previous work has shown that this preparatory perturbation can be linked to the optimal control solution to bring the cortical activity towards the unique motif-specific stable state instructed by an external input provided to cortex (Kao et al., 2020; Logiaco et al., 2019). However, the dynamics of the continuously nonlinear RNNs that we and modern ML practitioners consider (Maheswaranathan et al., 2019; Wyffels & Schrauwen, 2009; Sussillo & Abbott, 2009; Tani, 2003; Liu et al., 2019; Merel et al., 2019a) are richer and qualitatively different from this linear regime, and do not allow setting the parameters through analytics as done in the former neuroscience works (Kao et al., 2020; Logiaco et al., 2019). Indeed, in this limited linear regime, *(i)* single motifs could not be produced in response to a simple static input (Logiaco et al., 2019) – whereas the RNNs can (Fig. 2); *(ii)* spontaneous activity decays to zero, whereas units are spontaneously active in our networks (Fig. 3b, 'prior learning' curves, (Sompolinsky et al., 1988)); and *(iii)* a single stable fixed point exists, while continuously nonlinear RNNs can exhibit multistability (Maheswaranathan et al., 2019). Therefore, it is unclear whether such a preparatory connectivity perturbation can be successfully trained with gradient descent to shape a convergence phase in the more powerful performance-optimized continuously nonlinear RNNs. Also, to draw conclusions about the functionality and applicability of the preparatory connectivity perturbation, it is critical to determine whether it can act in synergy with gradient descent training of the motif-specific parameters in the continuously nonlinear RNNs. To answer these questions, we first design a training protocol for a preparatory connectivity perturbation (that we call 'preparatory module', Fig. 3a left). This training is done once and for all before learning any motif; details are in A.5.1 and we summarize the procedure below. We want to train the weights $\mathbf{U}_{\text{prep}}\mathbf{V}_{\text{prep}}^{\mathsf{T}}$ that can shape fast convergence to zero activity *in the absence of input* in nonlinear RNNs. To do so, we initialized the network activities with standard random initial $\mathbf{x}$ values and trained using the cost function $\sum_t \|\mathbf{r}(t)\|^2$ (Fig. 3a), while the dynamics of our networks obey:

$$\tau\dot{\mathbf{x}} = -\mathbf{x} + (g^x\mathbf{J} + \mathbf{U}_{\text{prep}}\mathbf{V}_{\text{prep}}^{\mathsf{T}})\tanh(\mathbf{x})$$

where $g^x$ is $g^{\text{ad}}$ or $g^{\text{mu}}$ for the additive and multiplicative networks respectively. We initialized the weights $\mathbf{U}_{\text{prep}}$ and $\mathbf{V}_{\text{prep}}$ with centered Gaussian with std $\sqrt{0.05/\sqrt{P*N}}$, with $P = 50$ (though smaller values of $P$ give similar results as long as $P/N$ is more than a few percent).

Then, with the thalamic preparatory modules in hand (whose weights are then fixed), we trained additive and multiplicative networks on individual motifs with the network activities $\mathbf{x}$ randomly initialized with a standard Gaussian, but starting each motif with a motif preparation period *now also involving a motif-specific input* (Fig. 3b). For the additive network, our only modification from before is that we included the thalamic preparatory module in the network dynamics for the first $5\tau$:

$$\tau\dot{\mathbf{x}} = -\mathbf{x} + (g^{\text{ad}}\mathbf{J} + \mathbb{1}_{t\leq 5\tau}\mathbf{U}_{\text{prep}}\mathbf{V}_{\text{prep}}^{\mathsf{T}})\tanh(\mathbf{x}) + \mathbf{b}_\mu.$$

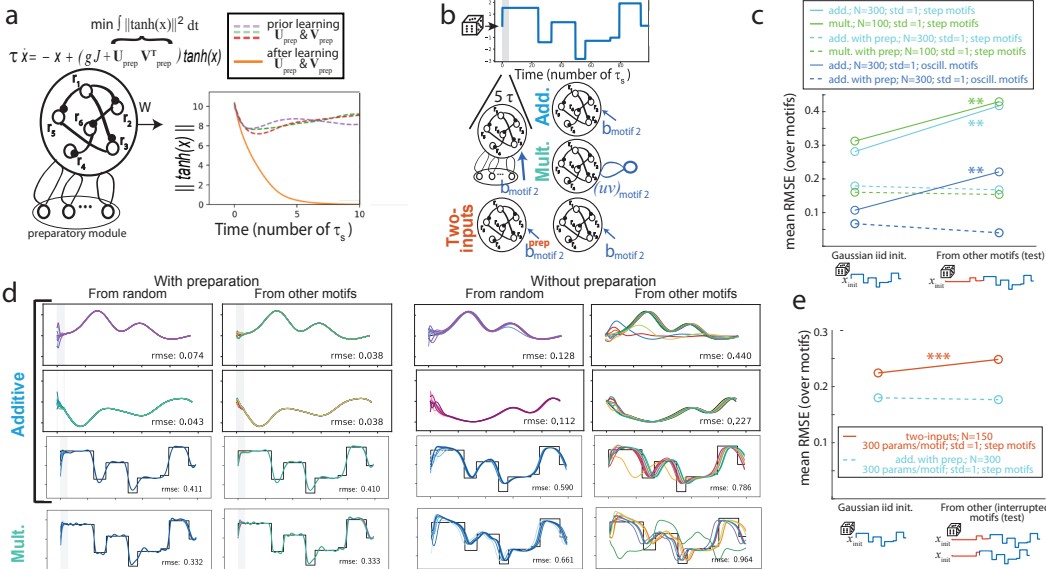

Figure 3: **Improvisation of motif sequences by RNNs with thalamocortical insights**.**a**) Training a preparatory module consisting of $P$ loops weights forming a perturbation $\mathbf{U}_{\text{prep}}\mathbf{V}_{\text{prep}}^{\intercal}$ of the connectivity, for a strongly nonlinear network. **b**) Training the motif-specific parameters in the additive, multiplicative and two-input architectures. **c**) Average root mean square error over motifs during training conditions vs. during transitioning, in the additive and multiplicative architectures; and with or without preparatory module. Ten step motifs are used. **d**) Example network performance when initializing the network state with the same random distribution as during training, as opposed to when transitioning from another motif. **e**) Same as c but comparing additive and two-inputs architectures using more motif transitions. Stars indicate a significant error increase (signed rank test).

For the multiplicative network, to make a more direct comparison with the thalamocortical model (Appendix A.4), we set the dynamics such that the input $\mathbf{b}_{\mu}$ and loop $\mathbf{u}_{\mu}\mathbf{v}_{\mu}^{\intercal}$ were only active during the preparatory and post-preparatory periods respectively:

$$\tau\dot{\mathbf{x}} = -\mathbf{x} + \left(g^{\text{mu}}\mathbf{J} + \mathbb{1}_{t\leq 5\tau}\mathbf{U}_{\text{prep}}\mathbf{V}_{\text{prep}}^{\intercal} + \mathbb{1}_{t>5\tau}\mathbf{u}_{\mu}\mathbf{v}_{\mu}^{\intercal}\right)\tanh(\mathbf{x}) + \mathbb{1}_{t\leq 5\tau}\mathbf{b}_{\mu}.$$

First, the RNNs trained with a preparatory module are much more accurate even during the production of single motifs initialized with the same distribution as used during training, especially for the more difficult step motifs ($\approx 50\%$, Fig. 3c and d, left). This strongly suggests that, when training with random initialization, the imposition of fast network dynamics by the preparatory module allows the tuned preparatory input to more efficiently steer the dynamics to explore and find a *motif-specific network state* that leads to more accurate motif production (Kemeth et al., 2021). Accordingly, we find that the preparatory module speeds up learning (see A.6.4). After training, the preparatory module also helps converging quickly to the correct motif-specific network state. Conversely, without the preparatory module, the motif-specific parameters appear to struggle to support accurate motif production on their own (create fast dynamics at motif start, shape the pattern towards which activity must converge, and modulate the shared RNN dynamics during motif production).

Second, the RNNs trained with a preparatory module are now able to tackle zero-shot transfer to chains of motifs at no performance cost, while driving *smooth interpolating transitions* (Fig.3c-e, Fig.14 top right). This strongly argues that the preparatory module shapes a wide attractive landscape at motif start that includes the complex correlated network states reached at the end of motifs.

Third, through nonlinear interactions between the preparatory input and the rich dynamics of the RNNs, our networks can match the motifs' shapes during preparation (Fig. 3d, first two rows). This is another advantage of our RNNs over the above-mentioned dynamically limited models (which can only implement a motif-independent interpolation during preparation, besides having more limited expressivity (Kao et al. (2020); Logiaco et al. (2019) and Appendix A.4). This enables robust, accurate and plausible motif transitions without any tuning of the duration of the preparatory period.

Finally, we want to stress the advantages of the preparatory connectivity perturbation $\mathbf{U}_{\text{prep}}\mathbf{V}_{\text{prep}}^{\mathsf{T}}$ compared to possible alternative approaches that would only try to match the apparent timecourse of the neural activity during preparation in the brain (Sussillo et al., 2015; Zimnik & Churchland, 2021). First, the preparatory loops can modulate the dynamics in a unit-specific way to create efficient convergence with a subtle modulation of the RNN connectivity (Fig. 12a,d). This is the reason why the preparatory input can still interact with the rich RNN's recurrent connectivity, which allows this input to shape the output into the desired motif even during preparation. Second, because these loops are not tuned for any particular motif, they can be optimized once and for all, and do not impose any additional cost when learning new motifs. Instead, when considering naive ways of setting specific dynamics at the beginnings of motifs such as using a second motif-specific input at the start of each motif ('two-inputs' approach, Fig. 3c), any parameter used for preparation scales with the number of motifs. We made this 'two-inputs' approach comparable to our preparatory module approach by applying the first input the network for the first $5\tau$ of the motifs and then a second input during the remainder of the motif, and by equalizing the number of parameters involved (as we had done between the additive and multiplicative network) by setting $N = 150$. We found that this strategy led to both decreased accuracy and robustness compared to using our preparatory module, with both effects combining to a large ($\approx 40\%$) increase in RMSE during transitioning. We could precisely assess the transitions robustness in these networks this by using an extended motif transition data set (Fig. 3e). More precisely, we considered transitions between pairs of step motifs where the transition occurs at fractions of 0.5, 0.6, 0.7, 0.8, 0.9, 1.0 of the total duration of the first motif (e.g. Fig. 3e right, bottom). This is a more challenging task – that involves a larger variety of network states from which to transition than before (then, we only considered the $100\%$ scenario where all motifs ended at the same value: zero). For these transitions, the 'two-inputs' network showed a significant increase in error – including occasional very large errors for some transitions, see Appendix A.5.2 – compared to initializing motifs with the randomness used during training. In contrast, the additive network had equally robust performance in both conditions (Fig. 3f). Thus, our results argue that using brain insights to constrain the architectures of networks has advantages, even beyond taking inspiration from the biological activity patterns (Li et al., 2019; Sinz et al., 2019; Zador, 2019).

## 5 DISCUSSION

We found that when trained on many randomly initialized single motifs, gradient-trained nonlinear RNNs struggle during zero-shot transfer to new sequence orders. This is relatively surprising given that such RNNs are known to be able to learn attractive trajectories (Laje & Buonomano, 2013; Sussillo & Abbott, 2009; Pollock & Jazayeri, 2020). This suggests that training mostly constrains the readout dimension in our setting. This is both a blessing - as it is the ultimate reason why different motifs can efficiently share parameters and thus benefit from a common dynamical baseline that is only partially modulated, without being hurt by irrelevant aspects of the dynamics (Logiaco et al., 2019; Russo et al., 2018) - and a curse as it makes flexible motif chaining difficult. In this context, we show that using a preparatory module whose architecture is brain-inspired and who imposes a dynamics that mimics motor cortical activity during hierarchical motor sequencing (Zimnik & Churchland, 2021) can efficiently enable zero-shot transfer to new sequence orders. Even more surprisingly, introducing the preparatory module in the nonlinear RNNs actually sizably improves single motif training such that the motifs are also more accurate during training conditions. These results reveal a general function for motor preparation specifically for flexible chaining of motor motifs, and show that it is not a byproduct of arbitrary biological hardware constraints or of the restricting assumptions made by previous models to allow for analytical tractability (Logiaco et al., 2019; Kao et al., 2020). Therefore, ultimately, for the application to robotics control - at least in a context where intrinsic dynamics matter (Liu et al., 2019; Yeo et al., 2016) - this preparatory module could be very useful (i) to increase accuracy, (ii) to enable smooth transitions governed by dynamics that can be tuned to avoid passing through undesirable states, and (iii) to offer a mechanism for learning appropriate transition times by gradient descent - which are all current challenges for state-of-the-art networks (Merel et al., 2019a;b). In addition, our approach is extendable to combining recurrent dynamics - useful in contexts where sensory feedback is not fully reliable (Yeo et al., 2016; Liu et al., 2019) - with corrections driven by sensory inputs (Guo et al., 2019). Our work therefore joins several recent calls outlining the need to infuse ANNs with more expert domain knowledge in order to achieve good generalization in real-world scenarios involving out-of-distribution generalization and zero-shot transfer (Russin et al., 2019; D'Amour et al., 2020; Zador, 2019).

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

# A APPENDIX

## A.1 FORMAL DEFINITION OF OUR PROBLEM SETTING

Here is how we formally define the task solved in our paper:

-Given a dynamical system $\mathcal{D}$ with (i) internal variables $\mathbf{h}$ (i.e., $\mathbf{h}(t)$ is defined through its derivative with respect to the continuous time variable $t$) and (ii) the property that $\mathcal{D}$ can be in one of $M$ different states indexed by $\mu$ (with $\mu \in 1 \dots M$, and $M$ is large and a priori unknown), each lasting a fixed duration $t_\mu$, such that whenever $\mathcal{D}$ is in state $\mu$, a function of the internal states $F(\mathbf{h})$ exactly follows a fixed continuous function $g_\mu$ (of length $t_\mu$) that we term a 'motif'. (Concretely, if a particular trajectory of states $[s_2, s_1, s_3]$ is visited, $F(\mathbf{h}(t))$ is formed of the concatenation of three continuous 'motifs' indexed by $[s_2, s_1, s_3]$.)

- Given a sequence of indices $I = [p_1, ...p_K]$ that label a sequence of states undergone by $\mathcal{D}$, where $\forall i, p_i \in \{1, \dots, M\}$

- Given labeled training examples that only contain a finite number $T_\mu$ of *discrete* samples from each continuous function $g_\mu$, and given that the learning from different $g_\mu$ can occur sequentially and extendably without interference

- Find a function $f$ which, for each function $g_k$, is parameterized by a vector $V_\mu$ of size $N_V < T_\mu$ (where $N_V$ is as small as possible) such that $f(I; , V_1, \dots, V_M, dt)$ produces a time series corresponding to $F(\mathbf{h})$ when $\mathcal{D}$ undergoes the sequence of states ordered by $I$, with an adjustable sampling interval given by $dt$ (such that a smaller $dt$ leads to a smoother time series).

Finally, we note that is also desirable that $f$ be noise-robust.

In our work, we solve this problem using an RNN to implement the function f – a natural choice given that it models a dynamical system $\mathcal{D}$. In addition, the tunable motif-specific parameters $V_\mu$ are either the input $\mathbf{b}_\mu$ for the additive network, or the concatenation of the input $\mathbf{b}_\mu$ and the loop parameters $\mathbf{u}_\mu$ and $\mathbf{v}_\mu$ in the multiplicative network. As we mention in the text, given that we work in a continual learning setting where we would like to learn an extendable library of many motifs, we are trying to limit the number of tuned parameters per motif $N_V$ (table 1). This is important for our networks to constitute viable solutions that can be built on for designing motor controllers, because it is then desirable to limit both the memory requirements and the time needed to update the state of the network.

Note that we ensured noise-robustness in our work by injecting noise during training in the hidden state of our RNN (which, by a simple transformation of variable, is equivalent to injecting a transformed noise into the input $\mathbf{b}_\mu$ received by the network). The larger component of the noise we used consists in a random initialization of the network state, and this noise then propagates to later timesteps through the recurrent dynamics (see section A.6.3).

## A.2 MOTIF GENERATION METHOD AND FULL LIST OF MOTIFS USED

For this study, we use two different types of motifs that challenge the dynamics of continuous time RNNs: oscillatory motifs, that are easier to produce; and step motifs that are harder.

### A.2.1 OSCILLATORY MOTIFS

To generate oscillatory motifs that would be relatively easy to produce by RNNs, we sampled the output from a random Gaussian network with $\tanh$ nonlinearity set in the chaotic regime (Sompolinsky et al., 1988; Sussillo & Abbott, 2009; Sussillo et al., 2015). To make these motifs very easy and possible to connect in 'seamless' chains, we also low-pass filtered the resulting trajectories.

Fig. 4 lists the oscillation motifs used in this paper.

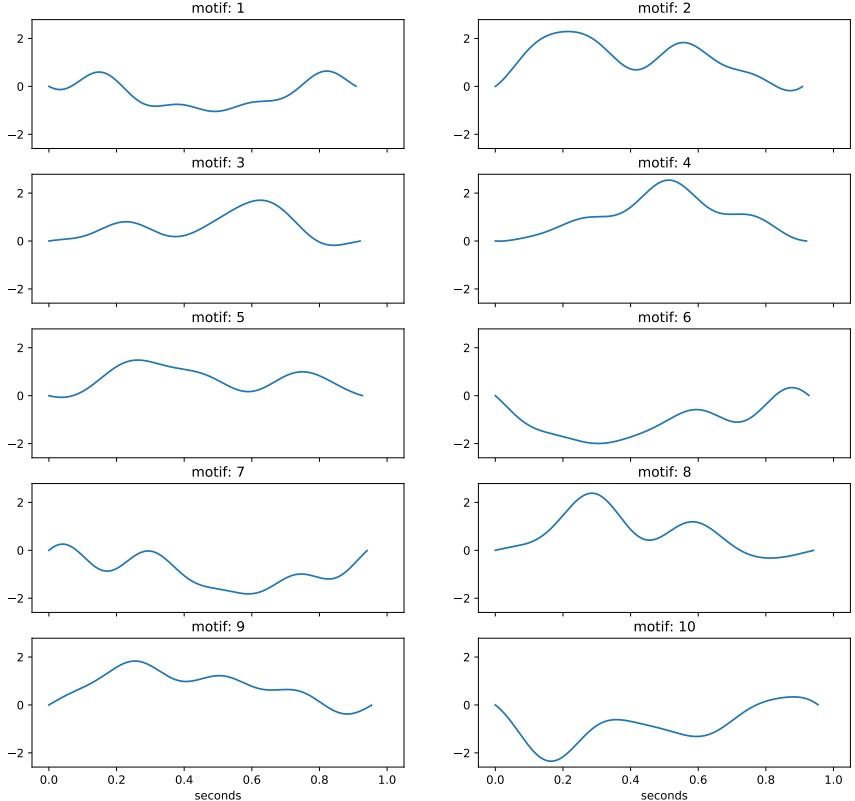

Figure 4: **Oscillatory motifs used in this paper**.

### A.2.2 STEP MOTIFS

These motifs consist of a series of positive and negative discrete jumps with intervening constant periods. To generate a particular motif, we follow the following steps. First, we generate a centered Ornstein–Uhlenbeck process $z$ by running the dynamics $dz/dt = -z + \left(\gamma/\sqrt{dt}\right)\chi(t)$, where $\chi(t)$ is taken from a standard Gaussian independently at each timestep, dt=0.1 and $\gamma = 3\sqrt{2}$ such that the steady-state standard deviation of this process is 3. After a 'warmup' period of $200\,dt$ that

we discard, we produce an additional $1000\,dt$ of z-values that will be the basis for generating the motif. Second, we draw random time intervals $T_k$ with a uniform probability between $50\,dt$ and $500\,dt$ and select the first few in a list $[T_1, T_2, ..., T_j]$ such that $\sum_{k=1}^{j} T_k < 1000\,dt$. Third, we set the value of the motif between the start and time $T_1$ to the average of the z-values over the same time interval; and similarly for the subsequent intervals, the value of the motif between the times $[T_{i-1} + dt, T_i]_{i \in [2,...,k]}$ is set to the average of the z-values over the corresponding interval. Finally, we pad the end of the motif with zeros for $50dt$ and also reset the first value of the motif to 0.

Fig. 5 lists the step motifs used in this paper.

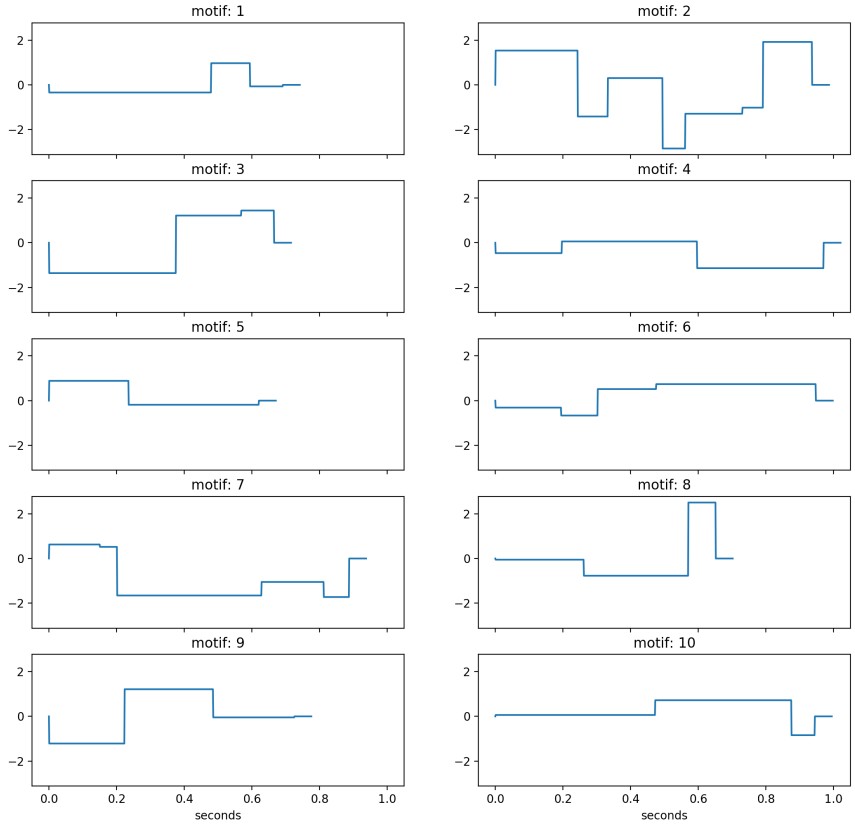

Figure 5: **Step motifs used in this paper**.

### A.3 Performance of 'vanilla' RNNs

Here we show examples for the motif-by-motif performance of different 'vanilla' RNNs (i.e., without a preparatory module) that relate to section 3 of the main text, where network training involves producing single motifs from a random initialization of the network state. The networks are tested both on new random initialization (in-distribution generalization) and during motif transitioning (out-of-distribution generalization).

#### A.3.1 Additive network, N=300, step motifs

In Fig. 6, we show an additive network of size 300 trained on the step motifs. This is a different network than the one we show in the main text Fig. 2a (i)-(iii) but the conclusions are identical –

despite the presence of a small variability between different networks trained from different initial seeds (see e.g. main text Fig. 1c).

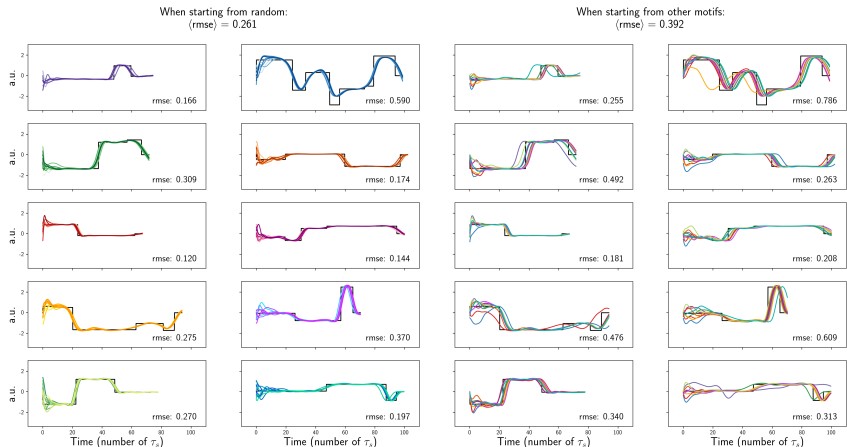

Figure 6: **Performance of an additive network, N=300, step motifs**. Left: the network outputs for each motif when starting from 9 different random **x** values. The saturations (light to dark) indicate different trials. Right: network outputs for each motif when starting from the **x** values taken at the end of the other 9 motifs. Colors indicate the identity of the prior motif (as labeled in the left panel). Numbers indicate the root mean square errors (rmse) for each motif (and their average over motifs).

### A.3.2 MULTIPLICATIVE NETWORK, N=100, STEP MOTIFS

In Fig. 7, we show a multiplicative network of size 100 trained on the step motifs. This is a different network than the one we show in the main text Fig. 2b but the conclusions are identical – despite the presence of a small variability between different networks trained from different initial seeds (see e.g. main text Fig. 1c). Note that the number of parameters tuned for motifs is adjusted relative to the additive network of size 300 shown previously (main text table 1).

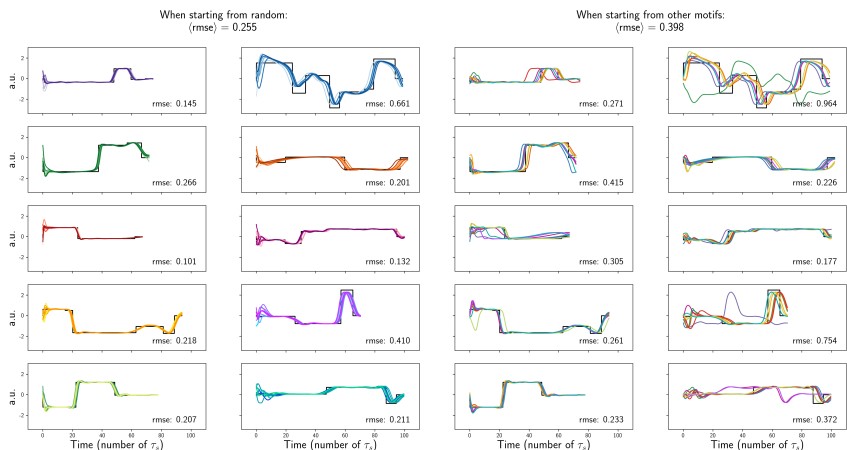

Figure 7: **Performance of a multiplicative network, N=100, step motifs**. Conventions as in Figure 6.

### A.3.3  ADDITIVE NETWORK, N=1000, STEP MOTIFS

In Fig. 8, we show an additive network of size 1000 trained on the step motifs. Notice that while increasing the network size improved the performance when starting from the random initialization used during training, there are still large errors when transitioning from other motifs.

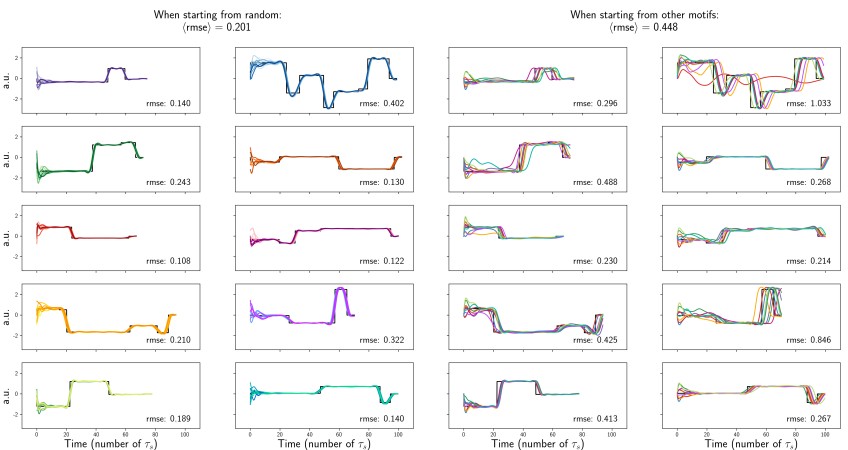

Figure 8: **Performance of an additive network, N=1000, step motifs**. Conventions as in Figure 6.

### A.3.4  ADDITIVE NETWORK, N=300, OSCILLATORY MOTIFS

In Fig. 9, we show an additive network of size 300 trained on the oscillatory motifs. Notice that while, with these easier motifs, the performance is really good when starting from the random initialization used during training (as compared to the performance for the step motifs), there are still large errors when transitioning from other motifs. Also, we want to stress that, during sequencing, even when focusing on a particular first motif, the network performance can be very different depending on the identity of the second motif. For instance, let's focus on the light green motif in the panel 'When starting from random', bottom row on the left - the sequences where this motif is the first motif are shown as green traces in the panel 'When starting from other motifs'. When this green light motif is the first motif in a sequence, the performance is almost flawless for following the motif in the panel 'When starting from other motifs', second line on the left. In contrast, the performance is very poor for another following motif situated in the panel 'When starting from other motifs', fourth line on the right.

### A.3.5  CONTROL NETWORK, N=50, STEP MOTIFS

In Fig. 10, we show a control network of size 50 trained on the step motifs. Like the additive and multiplicative networks, this control network struggles when generalizing from single motif training to motif sequencing.

### A.4  ANALYTICALLY TRACTABLE THALAMOCORTICAL MODEL

Here, we summarize the insights that can be gained from an analytically tractable thalamocortical model whose switching linear dynamics are constrained to stay in the linear regime during individual motif preparation and execution ((Logiaco et al., 2019), see also (Kao et al., 2020)). Importantly, the constraints on the dynamics of this model largely limit its computational capacity, and no online learning method has been suggested for this model so that the approach is strongly limited to simple scenarios where analytical tractability is possible. The work set forth in the main text is therefore critical to test and expand the ideas of this model in a setting that is amenable to solving useful tasks in an engineering sense, and for determining whether the biological features included in

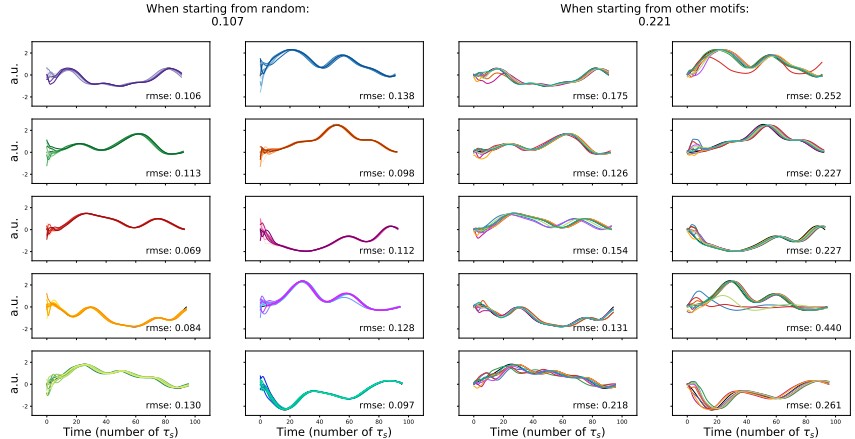

Figure 9: **Performance of an additive network, N=300, oscillatory motifs**. Conventions as in Figure 6.

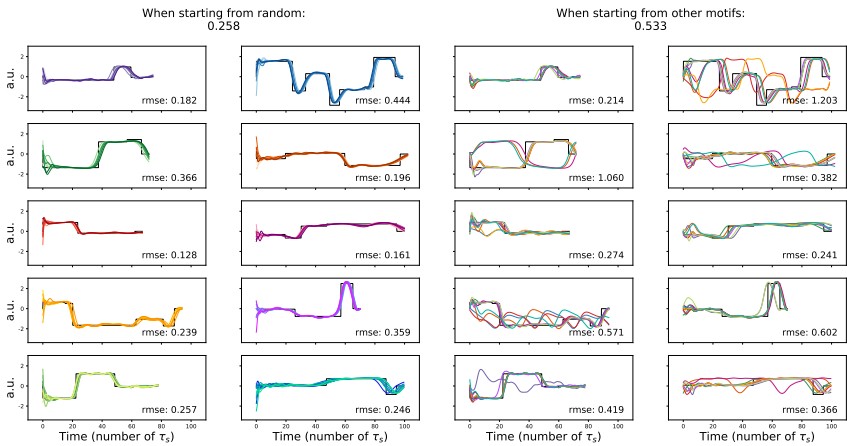

Figure 10: **Performance of a control network, N=50, step motifs**. Conventions as in Figure 6.

this model might have been selected as generally useful rather than reflecting random evolutionary idiosyncracies.

The model consists of a recurrent cortical module with connectivity $\mathbf{J}^{\mathrm{cc}}$ and activities $\mathbf{x}$ whose projection through the readout weights $\mathbf{w}$ constitutes the output. This cortical module interacts with a non-recurrent thalamic module through instantaneous loops consisting of corticothalamic and thalamocortical weights. The basal ganglia, which provides inhibitory input into thalamus, are modeled as selectively disinhibiting specific thalamic loops in order to cause execution of the associated motif.

**Motif execution:** During motif $\mu$, a single thalamic loop is disinhibited leading to the dynamics:

$$\tau \dot{\mathbf{x}} = \tilde{\mathbf{J}}_\mu \mathbf{x}, \quad \text{where} \quad \tilde{\mathbf{J}}_\mu \equiv g\left(\mathbf{J}^{\mathrm{cc}} - \mathbf{I}\right) + \mathbf{u}_\mu \mathbf{v}_\mu^\mathsf{T}, \tag{1}$$

with motif-specific loop vectors $\mathbf{u}_\mu$ and $\mathbf{v}_\mu^\mathsf{T}$. We now consider how these dynamics can approximate a desired output $y_\mu$, knowing that in general a good approximation for $y_\mu$ can be reached through a (preliminarily numerically identified) linear combination of a small number $K$ of complex exponentials: $y_\mu(t) \approx \hat{y}_\mu(t) = \sum_{k=1}^{K} [\hat{\boldsymbol{\alpha}}_\mu]_k e^{[\hat{\boldsymbol{\lambda}}_\mu]_k t}$ (see main text Fig. 1d; (Logiaco et al.,

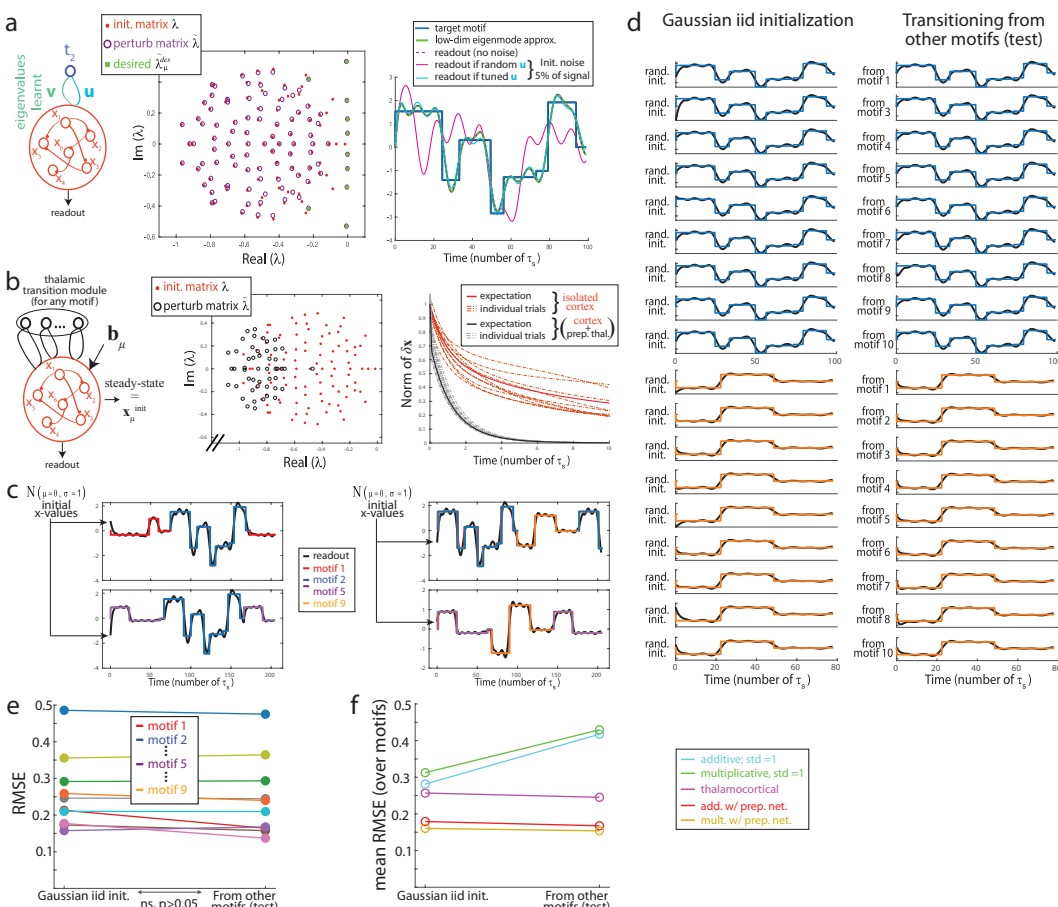

Figure 11: **Robust transitions in thalamocortical model**. **a**) Adjusting a motif-specific loop through the thalamic unit $\mathbf{t}_2$ (i.e. motor preparation, left), leading to the control of both eigenvalues (middle) and eigenvectors of the dynamics $\mathbf{x}(t)$ such that the readout robustly follows motif 2 (right). **b**) Thalamic module used for all motif transitions (which involves preparation of the cortical state to execute the next motif, left). After optimization, the thalamic module sets the eigenvalues of the dynamics to be more negative (middle) which results in a fast decrease of the distance to steady-state $|\delta\mathbf{x}|$ (right). **c**) Example sequences. **d**) Example trials starting from standard Gaussian random initial conditions vs. when transitioning from other motifs. **e**) Change of root mean square error for each motif between random initialization and zero-shot transfer to motifs transitions (not significant as per signed rank test). **f**) Comparing the performance between the thalamocortical model and the models presented in the main text, for both random initialization and zero-shot transfer to motifs transitions.

2019)). The cortical readout can exactly match $\hat{y}_\mu$ if the eigenvalues of $\tilde{\mathbf{J}}_\mu$ contain the entries of the vector $\hat{\boldsymbol{\lambda}}_\mu$ and if the initial network activities $\mathbf{x}_\mu^{\text{init}}$ are set correctly. We accomplish the former (Fig. 11a) by setting $\mathbf{v}_\mu = \mathbf{L}^\intercal \operatorname{diag}(\mathbf{L}\mathbf{u}_\mu)^{-1}\mathbf{Q}^+\mathbf{1}$, where $\mathbf{L}$ is the left eigenvector matrix of $\mathbf{J}^{\text{cc}}$, and $Q_{kj} = 1/([\hat{\boldsymbol{\lambda}}_\mu]_k - \lambda_j)$ where $\lambda_j$ is an eigenvalue of $g\,(\mathbf{J}^{\text{cc}} - \mathbf{I})$. Next, we set the initial activities at the beginning of motif $\mu$ to $\mathbf{x}_\mu^{\text{init}} = \tilde{\mathbf{R}}_\mu \operatorname{diag}(\tilde{\mathbf{R}}_\mu^\intercal\mathbf{w})^{-1}\boldsymbol{\alpha}_\mu$ where $\tilde{\mathbf{R}}_\mu$ contains right eigenvectors of $\tilde{\mathbf{J}}_\mu$ (with the first $K$ columns corresponding to the eigenvalues in $\hat{\boldsymbol{\lambda}}_\mu$), and $[\boldsymbol{\alpha}_\mu]_{k\leq K} = [\hat{\boldsymbol{\alpha}}_\mu]_k$ and $[\boldsymbol{\alpha}_\mu]_{k>K} = 0$.

The preceding two steps do not specify $\mathbf{u}_\mu$, and with random $\mathbf{u}_\mu$ the readout will be highly sensitive to noise in $\mathbf{x}_\mu^{\text{init}}$ (pink trace in Fig. 11a right; see (Logiaco et al., 2019)). However, if $\mathbf{u}_\mu$ is set to minimize the analytically-computed expected readout deviation due to noise in $\mathbf{x}_\mu^{\text{init}}$ by modify-

ing the eigenvectors of the dynamics, then robust readout is possible (cyan trace in Fig. 11a right; minimization of the cost $C(\mathbf{u})$ in (Logiaco et al., 2019)).

**Motif transitions:** To successfully transition to motif $\mu$, it is sufficient to implement a mechanism by which $\mathbf{x}$ approaches $\mathbf{x}_\mu^{\text{init}}$, which will be the case if the dynamics during a so-called "preparatory period" has $\mathbf{x}_\mu^{\text{init}}$ as its steady-state. Additionally, it is desirable that the transition dynamics are fast and that they do not cause large transients values on the readout while relaxing to steady-state (Kaufman et al., 2014). To achieve this, it is possible (Logiaco et al., 2019) to employ a specific thalamic subpopulation of size $P$ which is disinhibited during all motif transitions, as well as a constant input $\mathbf{c}_\mu$ specific to the upcoming motif $\mu$, leading to the dynamics:

$$\tau\dot{\mathbf{x}} = \mathbf{J}_{\text{prep}}\mathbf{x} + \mathbf{c}_\mu, \quad \text{where} \quad \mathbf{J}_{\text{prep}} \equiv g\left(\mathbf{J}^{\text{cc}} - \mathbf{I}\right) + \mathbf{U}_{\text{prep}}\mathbf{V}_{\text{prep}}^{\mathsf{T}}, \tag{2}$$

with $N \times P$ loop weights $\mathbf{U}_{\text{prep}}$ and $\mathbf{V}_{\text{prep}}$. With these dynamics, the activity at steady-state will match $\mathbf{x}_\mu^{\text{init}}$ if $\mathbf{c}_\mu = -\mathbf{J}_{\text{prep}}\mathbf{x}_\mu^{\text{init}}$ (Fig. 11b). Note that the difference $\delta\mathbf{x} = \mathbf{x} - \mathbf{x}_\mu^{\text{init}}$ between the cortical activities and their steady state decays at a rate that is independent of $\mathbf{c}_\mu$ and therefore of the upcoming motif: $\tau\dot{\delta\mathbf{x}} = \mathbf{J}_{\text{prep}}\delta\mathbf{x}$ for all $\mu$. This allows us to optimize the same weights $\mathbf{U}_{\text{prep}}$ and $\mathbf{V}_{\text{prep}}$ to favor rapid and smooth transitions between all pairs of motifs – even though this approach is limited: this optimization has to be motif-independent, so that the shape of the readout cannot be adjusted to the specific motif being prepared (while the continuously nonlinear RNNs can do this, see Fig. 3d). Following ref. (Logiaco et al., 2019), we achieve fast transitions by minimizing the time-integral of the expected square norm of $\delta\mathbf{x}$, with rates $\delta\mathbf{x}_0$ at the beginning of the transition period sampled iid. Here, we also augment our cost function with the time-integral of the expected squared derivative of the readout to favor smoother transitions on average. Our total cost function is therefore:

$$C(\mathbf{U}_{\text{prep}}, \mathbf{V}_{\text{prep}}) = E_{\delta\mathbf{x}_0}\left[\int_0^\infty dt\,\|\delta\mathbf{x}\|^2\right] + \beta\,N\,E_{\delta\mathbf{x}_0}\left[\int_0^\infty dt\left(\frac{d}{dt}\mathbf{w}^{\mathsf{T}}\delta\mathbf{x}\right)^2\right] \tag{3}$$

$$\propto \text{Tr}\left(\mathbf{R}_{\text{prep}}\left(\left(\mathbf{L}_{\text{prep}}\,\mathbf{L}_{\text{prep}}^{\mathsf{T}}\right) \odot \mathbf{\Lambda}\right)\mathbf{R}_{\text{prep}}^{\mathsf{T}}\right) + \beta\,N\,\mathbf{w}^{\mathsf{T}}\mathbf{R}_{\text{prep}}\left(\left(\mathbf{L}_{\text{prep}}\,\mathbf{L}_{\text{prep}}^{\mathsf{T}}\right) \odot \mathbf{\Gamma}\right)\mathbf{R}_{\text{prep}}^{\mathsf{T}}\mathbf{w}$$

where $\mathbf{R}_{\text{prep}}$ and $\mathbf{L}_{\text{prep}}$ are the right and left eigenvectors of $\mathbf{J}_{\text{prep}}$, and its eigenvalues $\boldsymbol{\lambda}^{\text{prep}}$ are used to compute $\Lambda_{ij} = -1/(\lambda_i^{\text{prep}} + \lambda_j^{\text{prep}})$ and $\Gamma_{ij} = \lambda_i^{\text{prep}}\lambda_j^{\text{prep}}\Lambda_{ij}$. Finally, $N$ is the number of cortical units and $\beta$ is a hyperparameter which trades off the relative importance of transition speed and readout smoothness.

Note that the model parameters can be adjusted through analytical and semi-analytical techniques which do not require stochastic gradient descent on the simulated dynamics. This is an advantage when strictly imposing linear dynamical regimes during each motor preparation and motor execution for the fixed type of autonomous computation that it was designed for, but these restrictions can be limiting to optimize the network for more complex objectives (for instance, including correcting responses to sensory feedback).

We simulated the model on the same task as the gradient-trained RNNs, using our step motifs. The motif-specific parameters scale as in the multiplicative architecture (main text table 1), but the thalamocortical model does have a few more hyperparameters. To make sure that these did not induce an inability to compare between approaches, we reduced the cortical size to $N = 99$. Further, after exploring a few values, we set $g/\tau = 0.5$, $K = 10$, $P = 50$, $\beta = 1/20$, and the motif transition duration to $5\tau$. The readout weights $\mathbf{w}$ and recurrent weights $\mathbf{J}^{\text{cc}}$ were sampled from a centered Gaussian distribution with std $1/\sqrt{N}$. The approximations $\hat{y}_\mu$ were fit to the target motifs under the constraints that $\hat{y}_\mu(0) = 0$, that the elements of $\hat{\boldsymbol{\lambda}}_\mu$ had negative real part and were at least $\epsilon = 0.05$ apart from each other, and that the magnitudes of the elements of $\hat{\boldsymbol{\alpha}}_\mu$ were not exceedingly large (no larger than 3). The resulting $\hat{\boldsymbol{\lambda}}_\mu$ and $\hat{\boldsymbol{\alpha}}_\mu$ were then used to optimize $\mathbf{u}_\mu$, $\mathbf{v}_\mu$, and $\mathbf{c}_\mu$ as described above.

We generated sequences from the thalamocortical model after initializing with iid standard Gaussian samples for the elements of $\mathbf{x}$ (this choice of a unit standard deviation indeed leads to readout values within the same range as the target motifs, Fig. 11c). Interestingly, the motifs were produced with the same reliability when preceded by another motif in a sequence or when starting from random initial conditions (Fig. 11d,e). Also, choosing $K = 10$ pushed the network close to its limit in terms of noise robustness, without allowing the constrained thalamocortical network to reach the performance of nonlinear RNNs with the preparatory module (Fig. 11f).

In conclusion, the constrained thalamocortical model is therefore limited in its expressivity, in its capacity to shape the readout during motif transitions, as well as by the rigid and unpractical procedure by which its weights are adjusted.

We will now present additional results demonstrating that insights from this constrained thalamocortical model can be weaved into RNNs to combine zero-shot transfer ability with large expressivity and the flexibility afforded by gradient-based training.

### A.5 WEAVING IN INSIGHTS FROM THE THALAMOCORTICAL MODEL IN GRADIENT-TRAINED ANNS

In Section 4 of the main text, we presented the results of augmenting each of the additive and multiplicative networks with a 50-unit thalamic transition module that is active for the first $5\tau$ of every motif. Here we present more detail about these results.

#### A.5.1 DYNAMICS OF THE RNNS INTERACTING WITH A THALAMIC TRANSITION MODULE

The dynamics of our nonlinear recurrent networks with the transition module but without any input are:

$$\tau \dot{\mathbf{x}} = -\mathbf{x} + (g^x \mathbf{J} + \mathbf{U}_{\text{prep}} \mathbf{V}_{\text{prep}}^{\mathsf{T}}) \tanh(\mathbf{x}),$$

where $g^x$ is $g^{\text{ad}}$ or $g^{\text{mu}}$ for the additive and multiplicative networks respectively. The weights in $\mathbf{U}_{\text{prep}}$ and $\mathbf{V}_{\text{prep}}$ were initialized with centered Gaussian with std $\sqrt{0.05/\sqrt{P*N}}$. We trained the weights of the transition modules with ADAM under the cost function $\sum_t |\mathbf{r}(t)|^2$ where $\mathbf{r}(t) = \tanh(\mathbf{x}(t))$. Minibatches consisted of 64 trials of length $20\tau$, each starting with random $\mathbf{x}$ values sampled iid from the standard normal distribution. After 1,000 minibatches, both $N = 300$ and $N = 100$ networks were seen to have converged.

Figs. 12a,d show the eigenvalue distributions of $g^x \mathbf{J} + \mathbf{U}_{\text{prep}} \mathbf{V}_{\text{prep}}^{\mathsf{T}}$ after training. Importantly, though the thalamic module is low rank ($P < N$), all eigenvalues have real part significantly less than 1 which causes decay of the network rates towards a $\mathbf{0}$ fixed-point in the vicinity of this fixed-point. For comparison, we show the eigenspectrum of $g^x \mathbf{J}_{\text{rnd}} + \mathbf{U}_{\text{prep}} \mathbf{V}_{\text{prep}}^{\mathsf{T}}$ for random matrix $\mathbf{J}_{\text{rnd}}$ which has the same statistics as $\mathbf{J}$. In this case, there continues to be large eigenvalues that will prevent fast rate decay. These results demonstrate that the solutions $\mathbf{U}_{\text{prep}}$ and $\mathbf{V}_{\text{prep}}$ needed to negate the amplifying dynamics of $\mathbf{J}$ are specific to that particular $\mathbf{J}$.

In Figs. 12b,e, we show the time evolution of the norm of the rate vector $\mathbf{r}$ during three sample trajectories prior to learning and one post-learning (all post-learning samples are nearly identical). On the scale plotted, the norms after learning are indistinguishable from zero after approximately $7\tau$.

With the thalamic transition modules in hand, we retrained the additive and multiplicative networks as described in the main text Sections 2 and 3. For the additive network, our only modification from before is that we included the thalamic transition module in the network dynamics for the first $5\tau$:

$$\tau \dot{\mathbf{x}} = -\mathbf{x} + (g^{\text{ad}} \mathbf{J} + \mathbb{1}_{t \leq 5\tau} \mathbf{U}_{\text{prep}} \mathbf{V}_{\text{prep}}^{\mathsf{T}}) \tanh(\mathbf{x}) + \mathbf{b}_\mu.$$

For the multiplicative network, to make a more direct comparison with the thalamocortical model presented in the previous section of the Appendix, we set the dynamics such that the input $\mathbf{b}_\mu$ and loop $\mathbf{u}_\mu \mathbf{v}_\mu^{\mathsf{T}}$ were only active during the transition and post-transition periods respectively:

$$\tau \dot{\mathbf{x}} = -\mathbf{x} + (g^{\text{mu}} \mathbf{J} + \mathbb{1}_{t \leq 5\tau} \mathbf{U}_{\text{prep}} \mathbf{V}_{\text{prep}}^{\mathsf{T}} + \mathbb{1}_{t > 5\tau} \mathbf{u}_\mu \mathbf{v}_\mu^{\mathsf{T}}) \tanh(\mathbf{x}) + \mathbb{1}_{t \leq 5\tau} \mathbf{b}_\mu.$$

Figs. 12c,f show that both networks learn to perform the task and show no degradation in their performance when tested on sequence generation (i.e., having initial $\mathbf{x}$ values given by the ends of other motifs) rather than when starting them with standard Gaussian $\mathbf{x}$ values (which was their training regime).

Similarly, Fig. 13 shows that training on smooth motifs using a preparatory module leads to robust performance during zero-shot transfer to motif chaining in an example additive network (compare to Fig. 9). Notice how the shape of the motif is also matched during the preparatory period, which is enabled by the interaction between the motif-specific input $\mathbf{b}_\mu$ and the nonlinear $\tanh$ dynamics.

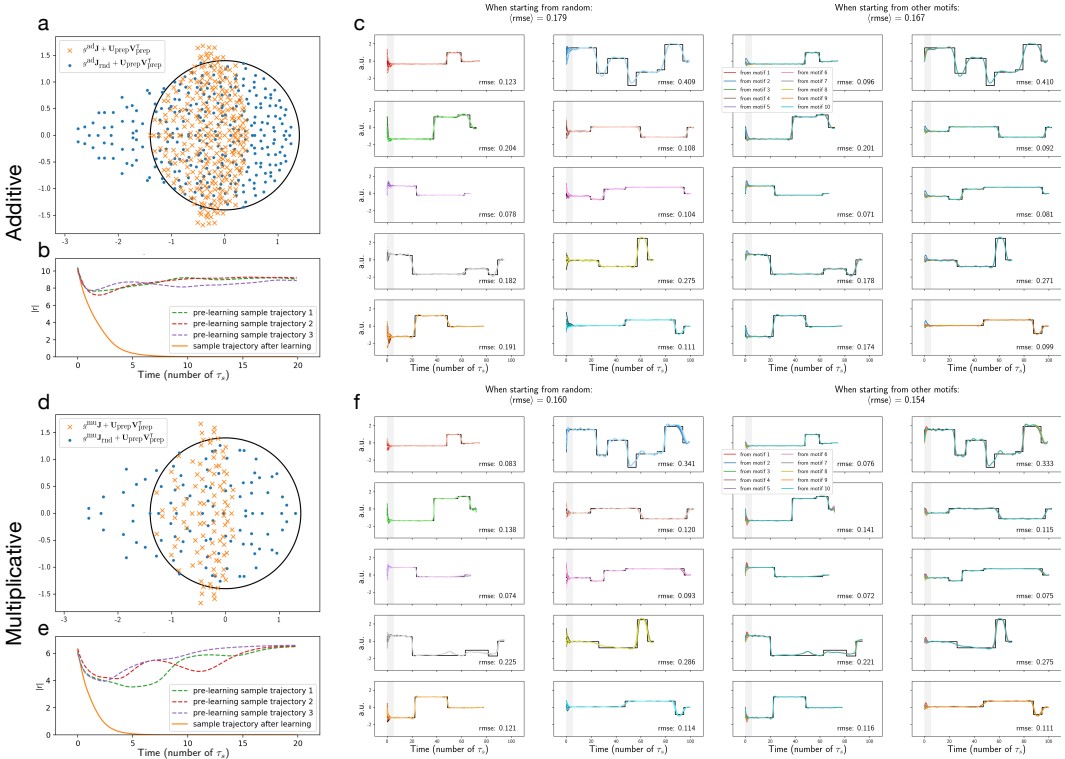

Figure 12: **Using a thalamic transition module rescues transitioning for nonlinear RNNs trained with SGD**. **a,b,c** Additive model. **a**. Eigenspectrum of $g^{ad}\mathbf{J} + \mathbf{U}_{prep}\mathbf{V}_{prep}^{\mathsf{T}}$ after training of $\mathbf{U}_{prep}$ and $\mathbf{V}_{prep}$ (orange crosses) and when replacing $\mathbf{J}$ with a random matrix not used during training (blue dots). Black circle has radius $g^{ad}$. **b**. $|\mathbf{r}|$ versus time before (dotted lines) and after (solid line) training of $\mathbf{U}_{prep}$ and $\mathbf{V}_{prep}$. **c**. The grey bars indicate the time during which the transition module was active. Other conventions as in Figure 6. **d,e,f**. As in **a,b,c** for the multiplicative model.

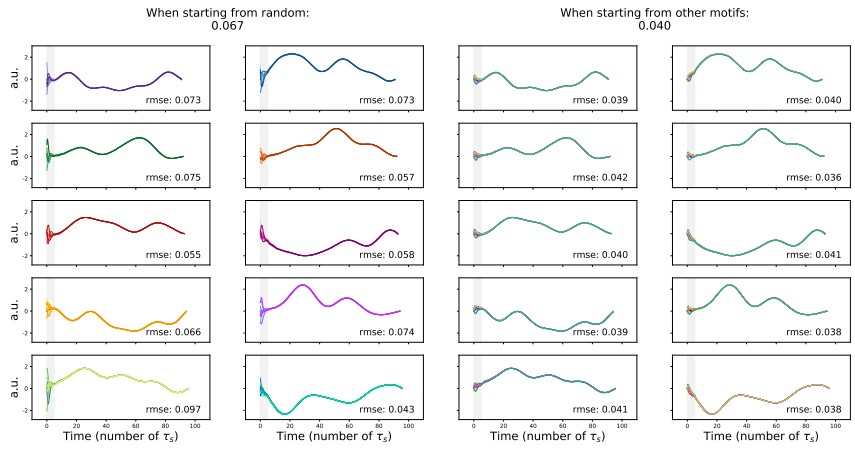

Figure 13: **Using a thalamic transition module rescues transitioning between oscillatory motifs in an additive ANN**. Conventions as in Figure 6.

### A.5.2 COMPARING THE THALAMIC TRANSITION NETWORK TO OTHER WAYS OF IMPLEMENTING A DIFFERENT DYNAMICAL REGIME AT TRANSITIONS

In the main text, at the end of section 4, we gave statistics about a comparison between using transition module described above and an alternative approach for setting the RNN in a different dynamical state at the beginning of a motif. More specifically, we considered a 'two-inputs' approach where we use a first motif-specific input during the first $5\tau$ of each motif to try to facilitate transitions, and a second motif-specific input during the rest of the motif to facilitate the production of the motif by the network.

Following the general approach in our article which adjusts the number of parameters tuned per motif (main text table 1), we set the size of the recurrent network to $N = 150$ so that the two inputs together lead to 300 parameters tuned per motif (and therefore 3000 parameters tuned in total when learning the ten motifs).

We used an extended data set consisting of transitions between pairs of step motifs where the transition occurs at 50%, 60%, 70%, 80%, 90%, or 100% of the total duration of the first motif. This is a more challenging task than before when we only considered the 100% scenario because all motifs end at the same value (zero). As stated in the main text, we found that the 'two-input' network did not perform as well as the matched additive network with a transition module (as described in the previous subsection of the Appendix, this module is not tuned to particular motifs, so the size of the recurrent network is here simply $N = 300$ to match the per-motif parameter count with the 'two-input' network).

In Fig. 14, we show examples for the single-motif performance of the 'two-input' network and the matched additive network, when transitioning from 70% vs. 100% of the duration of the previous motif (the latter one being the same situation as considered elsewhere in this work).

Examining Fig. 14 reveals that the 'two-inputs' network behaves differently from the additive network in two major ways. First, its overall motif accuracy is lower; and second, when transitioning from 70% of the duration of the previous motif, their accuracy consistently decreased compared to when initializing the motif at random as during training. This decrease in accuracy was seen consistently when transitioning in all scenarios when the previous motifs interrupted before their ending (which is zero-padded). This is probably because the larger variability of network outputs when interrupting motifs is reflected in a larger variability of network states. Importantly, the additive network with a transition module could handle this variability to allow the network to produce a more accurate output when chaining an interrupted motif with a subsequent one.

These results strongly suggest that 'engineering' a fixed point in the dynamics at the start of motifs (at a position that is adjusted with a preparatory input) is a particularly potent way of ensuring accurate and robust motif performance. In addition, close to a fixed point (which is presumably a good description of the dynamical regime close to the end of the preparatory period), the dynamics is quasi-linear, in which case classical results from control theory indicate that feedback loops such as those of the transition module are the optimal strategy to steer the network towards its target. Therefore, these results further strengthen our conclusion that the transition module we propose is a particularly potent strategy for zero-shot transfer during motif chaining.

## A.6 NOTES ON TRAINING

### A.6.1 ADAM PARAMETERS

We tested various parameters of ADAM. We identified the following as yielding successful training in our setting: learning rate $= 10^{-3} - 10^{-4}$, $\beta_1 = \beta_2 = 0.5$, and $\epsilon = 10^{-8}$.

Default parameters for ADAM most often worked very well (especially for the additive networks), but the parameters above could help training for the small control networks, or occasionally for the multiplicative networks – whose training was often non-monotonous (see Fig. 15 below), which could be especially limiting when training easier motifs for which the errors are small and/or the network activity varies little between different runs.

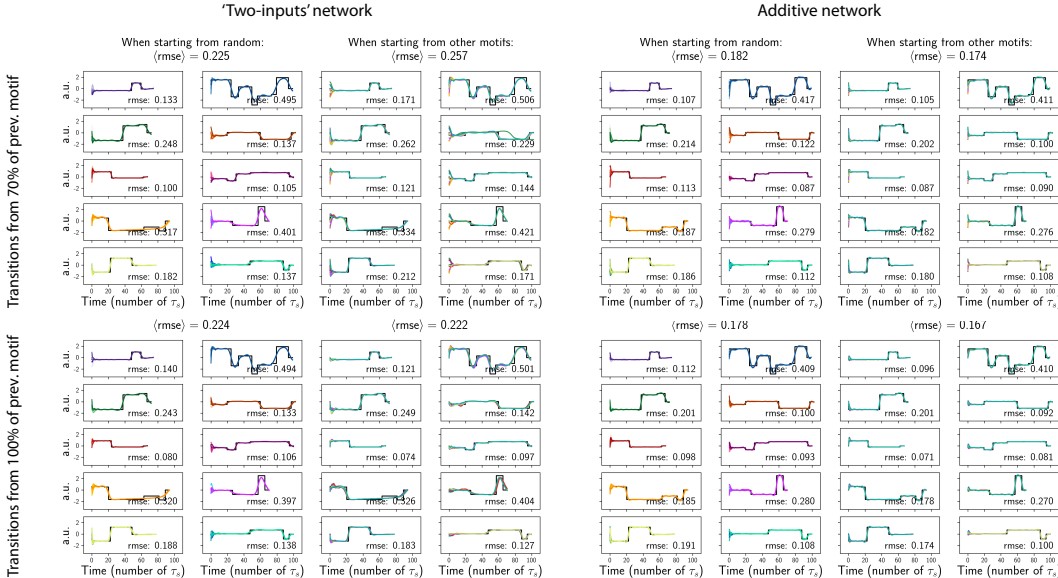

Figure 14: **Comparing a 'two-inputs' vs. additive network**. Left half: 'two-inputs' network; right half: additive network, adjusted for the number of parameters tuned per motif. Top half: comparing random initialization as experienced during training, and transitioning from 70% of the duration of other motifs. As a reminder, in the panels 'when starting from other motifs', colors indicate the identity of the prior motif (as labeled in the panel 'when starting from random' - where the motifs in this panel are shown in full, even though here during transitioning they were interrupted at 70% of the duration shown). Notice that, during sequencing, the networks smoothly interpolate between the various output values at the moment when the first motif is interrupted, and the readout value at the start of the next motif, without having to make a detour through a fixed readout value (as would occur if the networks were reset to a fixed – i.e., non motif-specific – state before each motif). In the additive network, this is because the preparatory module directly drives the RNN from its state at the moment when the first motif is interrupted, to a state specific to the second motif (and instructed by the motif-specific preparatory input) that corresponds to the readout at the end of the preparatory period. Bottom half: comparing random initialization as experienced during training, and transitioning from 100% of the duration of other motifs. For each quadrant, the two left vs. two right columns differ in the initial state of the motif with conventions as in Figure 6.

### A.6.2 DIFFERENCES IN LEARNING CURVES BETWEEN THE ADDITIVE AND THE MULTIPLICATIVE NETWORKS

Our investigation through hyperparameter search of the performance of additive and mutliplicative networks matched in number of tuned parameters (by fixing the size of the additive network to $N = 300$ and the size of the multiplicative network to $N = 100$) suggests that they lead to similar errors on average when trained to produce the step motifs (main text Fig.1c). Interestingly, we observed that these similar performances were typically reached through different types of learning curves (Fig. 15): the multiplicative networks tend to show discontinuous steps in the learning curve, whereas the additive networks typically have smooth learning curves. This suggests that the performance of the multiplicative networks may be limited by the ability of gradient descent to fully optimize these networks to their best possible performance. In the future, it would be interesting to investigate ways of improving the optimization of the multiplicative networks - for instance using motif-specific learning rate schedules, or by increasing its size to $N = 300$ while only training one-third of the weights of the loop and input vectors (so that it has as many untuned parameters as the additive network).

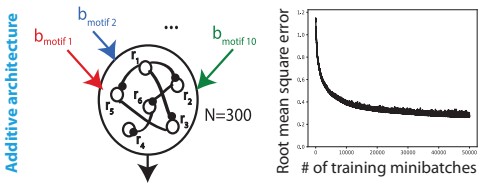 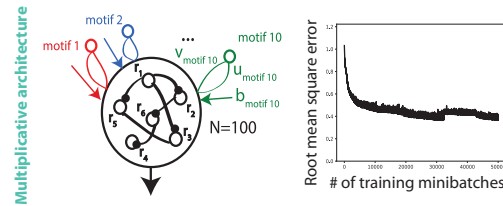

Figure 15: **Representative example learning curves for the additive and multiplicative networks (without a preparatory module)**. These are chosen examples from the networks shown in the main text Fig.1c. In this case, because we were trying to compare the small control network (which was harder to train) with the additive and multiplicative networks, we used the non-default ADAM parameters described in the previous section with a small learning rate (learning rate $= 10^{-4}$) Note that all transitions are included in the training set here.

### A.6.3 ADDING NOISE DURING TRAINING IS NECESSARY TO GET NOISE-ROBUST NETWORKS

The RNNs we propose need to be noise robust so that they can constitute viable solutions that can be built upon for motor control in the real world.

First, a transition to a new motor motif can be triggered in order to correct a corrupted motor command that corresponds to a noisy network state.

Second, for more general applications of the current framework to motor control in the real world, the dynamics often need to be modulated by sensory input – which is never perfectly reliable. In particular, the triggering of motif transitions is likely to often depend on sensory input.

For these reasons, we imposed noise robustness through a relatively large noise at motif start, implemented through a random initialization of the state of the network: at each run, the initial activities $x$ are sampled iid from a Gaussian distribution. Importantly, because of the recurrent dynamics, the initial noise is also propagating at later times through the recurrent dynamics. Additionally, in most of our simulations - including all the simulations with our preparatory module - we added a small (Gaussian with $std = 0.001$) amount of noise to the dynamics at each timestep.

Here, we show that the inclusion of noise is indeed necessary for getting noise-robust solutions (Fig. 16), illustrating this for additive networks. When training networks on single motifs initialized from the origin without noise - as opposed to adding Gaussian noise to the origin as we did in the main text - testing the networks with small amounts of noise leads to unreliable network outputs (Fig. 16 a, second column). In contrast, a network trained on motifs initialized from a standard Gaussian as a more reliable output across a range of standard deviation of the initialization (Fig. 16 b, first three columns). Finally, we note that while training the vanilla additive network without noise yields a higher accuracy in the noiseless conditions compared to a vanilla RNN handicapped by the requirement of noise robustness during training (Fig. 16 a left vs. Fig. 16 b left), even in these noiseless initialization conditions the vanilla additive network does not perform better than an additive network with a preparatory period and a noisy initialization (Fig. 16 a left vs Fig. 12c).

Therefore, not only do our biologically-inspired networks with a preparatory period display robust generalization to both in-distribution and out-of-distribution perturbations, they also do it at no accuracy cost compared to networks that are much less noise robust, and at no additional cost in terms of number of tuned parameters per motif. In addition, we will see in the following section that our preparatory module also speeds up learning.

### A.6.4 THE PREPARATORY MODULE COMBINED WITH RANDOM INITIALIZATION SPEEDS UP LEARNING

In the last section of the main text results, we argued that the preparatory module - which introduces fast convergence timescales at the beginning of motifs - helps single motif training. Indeed, when initializing motifs from a standard Gaussian both during training and testing, we observed that the networks with a preparatory module reach much lower error than the networks without such a module (Fig. 3c left, dashed lines vs. full lines).

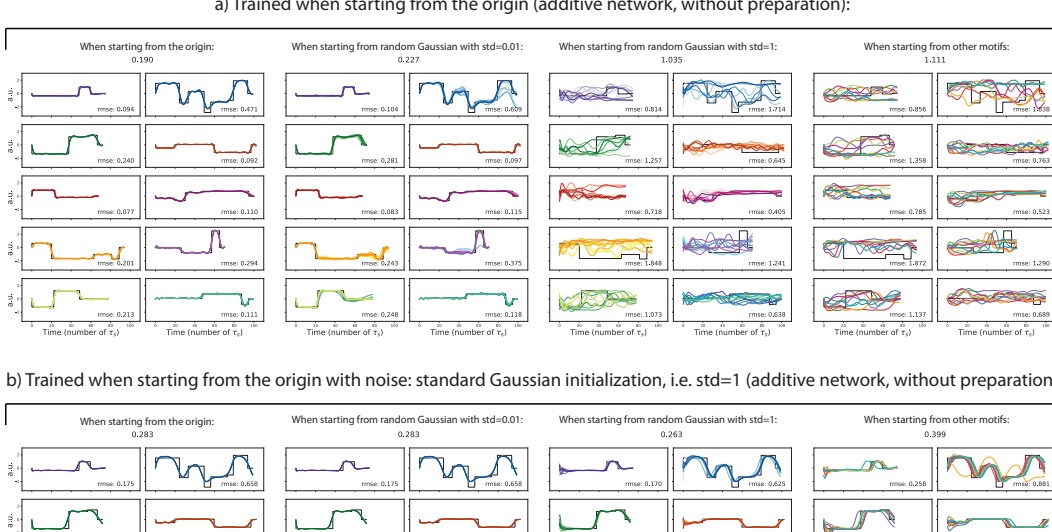

Figure 16: **Adding noise in the initial conditions during training improves noise robustness.** Comparing two additive networks (without preparation) when trained on single motifs that are: **a**) initialized at the origin without noise; **b**) initialized with standard random Gaussian values (standard deviation of 1), i.e. around a noisy origin. Both networks also receive a small additive noise at each timestep (drawn iid from a Gaussian with standard deviation of 0.001). The first three columns show the testing performance, when varying the standard deviation of the random initial conditions used to initialize motif production (centered iid Gaussian, from left to right: no noise, i.e. std=0; very small noise with std=0.01; standard: std=1). Last column: testing performance when motifs are initialized from the activity of a previous motif during two-motif sequences (here, the first motif is initialized from the origin without noise). Notice how training without noise leads to strong noise sensitivity (in **a**, some trials are very inaccurate even in the very small noise condition), whereas the networks trained when initializing motifs from a standard Gaussian do not lead to 'catastrophic' trials across all standard deviations of the random initialization (**b**, first three columns). Note that while training the vanilla additive network without noise yields a higher accuracy in the noiseless conditions compared to a vanilla RNN handicapped by the requirement of noise robustness during training (**a** left vs. **b** left), even in these noiseless initialization conditions the vanilla additive network does not perform better than an additive network with a preparatory period with a noisy initialization (**a** left vs. Fig. 12c). Color conventions as in Fig. 6. Standard ADAM parameters with a learning rate of $10^{-3}$).

Here, we give more evidence of this by examining the learning dynamics of the networks with or without a preparatory module. We find that networks trained with a preparatory module and standard Gaussian initialization reach a performance plateau faster (Fig. 17). Interestingly, not only is this true compared to training without a preparatory module with the same a standard Gaussian initialization (Fig. 17 a middle and right; and Fig. 17 b), but is is also true when compared to training without a preparatory module from a noiseless origin (Fig. 17 a left).

Taken together, our results strongly argue that our preparatory module helps network performance in two ways. First, when training single motifs from a fixed noisy initialization, the fast dynamics timescales introduced by the preparatory module appear to help the training-induced changes in the input to drive the exploration of different possible activity states until finding a motif-specific one that leads to particularly accurate motif production (Mohajerin & Waslander, 2019; Kemeth et al., 2021). Second, it helps filtering out noise and robustly converging to this motif-specific activity

pattern to start motif production. Therefore, the preparatory module helps the network implement computations that are shared between motifs, and then acts as an inductive bias such that the motif-specific parameters can be fully devoted to motif-specific functions (shape the motif-specific readout and final activity pattern during the preparatory period, and modulate the network dynamics during the remainder of motif production for accurate readout). This is why the preparatory module can be trained once and for all before training any motif specific parameters, which leads to improvements in accuracy and robustness at no per-motif cost in terms of tuned parameters and without introducing any learning interference in a continual learning scenario.

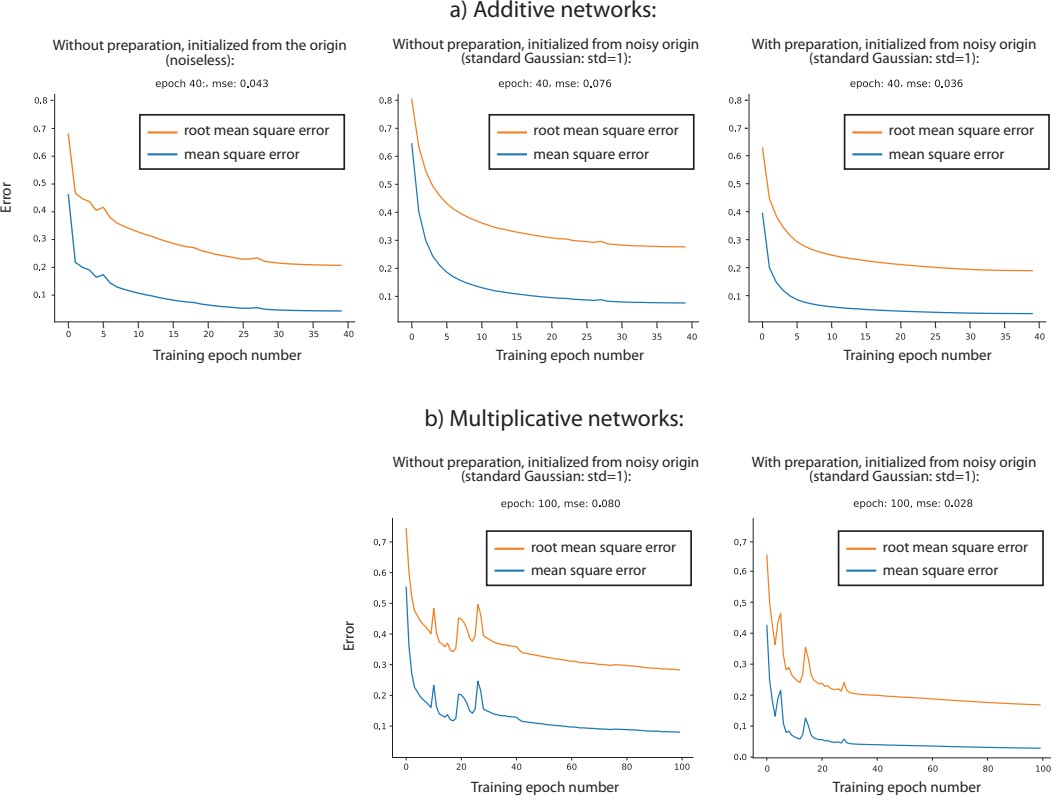

Figure 17: **Using a preparatory module along with a random initialization during training speeds up learning**. Evolution of the error (the mean square error - our objective function - in blue, and its square root in orange) over training epochs (each of these consists of several training batches), when training networks on single step motifs. Standard ADAM parameters with a learning rate of $10^{-3}$ are used, and a small Gaussian iid noise ($std = 0.001$) is added to the dynamics at each timepoint. **a**) Additive networks, without a preparatory module with either noiseless initialization (left) of standard Gaussian initialization (middle); or with a preparatory module and standard Gaussian iid initialization of motifs (right). **b**) Multiplicative networks, with standard Gaussian initialization, either without (left) or with (right) a preparatory module. The preparatory module speeds up learning in both additive and multiplicative networks.

