# OpenReview forum: "Brain insights improve RNNs' accuracy and robustness for hierarchical control of continually learned autonomous motor motifs"
_ICLR.cc/2022/Conference — ICLR 2022 Submitted_

### Official Review · Reviewer_W1np · 2021-11-02

**Correctness:** 2
**Technical Novelty And Significance:** 2
**Empirical Novelty And Significance:** 2
**Recommendation:** 3
**Confidence:** 3

**Main Review:**

**Strengths**

Continual learning is specially relevant as a setting for the purposes of this paper, partly given that learning from non-stationary distributions is closer in many aspects to the way humans learn. As such, the problem of catastrophic forgetting and generalization in continual learning are important research topics that are quite relevant for this venue. In this sense, the study of some of the architectures presented here (e.g. inspired on Schuessler et al. 2020) are interesting. Also welcomed are explicit addressing of the continuous time problem, which is relevant for many fields including control. Finally, the attempt to connect with Neuroscience by introducing a model inspired by thalamocortical interactions with the preparatory module is also acknowledged.

**Weaknesses**

- **Technical**: At the technical level, I found a couple of issues that affects the task described in section 1. First, there are references to the continual learning setup, but I can’t find the specifics, such as figures and detailed comments about the task structure etc., apart from mentioning the inclusion of all the motifs vs a sequential presentation (for which results are not shown). If the models are to be analyzed in this setting, it is important to see a robust description of the task and the metrics (see, for example, Chaudhry et al. 2019). Also, although I recognize the value of synthetic tasks, I also would have preferred to see in addition a task that encodes a “naturalistic” dataset, specially given that the authors are interested in a general noise-robust solution.

	Second, there is a problem that the authors seems to be well aware of as per the second to last paragraph in section 3. If I understood it correctly, the task described in section 2 admits a trivial solution, that of a hard resetting to an initialization state before processing a motif. This raises some considerations, including importantly that point (i) in that paragraph should be demonstrated empirically, regarding to different possible initialization schemes.


- **Writing** In my opinion this article suffers a series of important defects in its writing that ultimately have a strong negative impact in the communication of its scientific ideas. Please note that the intended feedback here is not about writing style. We acknowledge that not all papers are meant to be written the same way. However, I found reading this paper to be somewhat arduous and the presentation confusing. It also contains many questionable decisions into what constitutes the main text versus what goes in the appendix (e.g. presenting the numeric values of the hyperparameters for Adam in the first paragraph of the task description, while no formal description of the problem is included in the main text).

	Being constructive, I think this situation could be alleviated by introducing a better formalization of the task and the optimization problem in a simpler manner.
	- *Description of the problem*:
	The description of the problem in section A.1 could be improved by stating the optimization problem in simple terms. This can be done collecting information that is scattered in section 2, and introducing the statements with a notation consistent with the rest of the paper (I found strange that uses other notation from the main text). Please note that in section A.1 some terms are not defined (“descriptor vectors”) or introduced but seemingly not used later (N_V is stated to be as small as possible). Also, no discretization method is specified, for example, and the information about initialization is scattered.

		A beautiful example in this domain that you cite is Schuessler et al. (2020), which targets a similar conference in section 2: the equation of evolution of the RNN seen as a dynamical system (e.g. what you have in your first equation), followed by a clear specification of each of the terms (input, output, parameters, target function, etc.), parameters specific for each task, details about training and initialization, etc.

	- *Description of the task*:
	Section 2 in the paper is devoted to the description of the task and introduces the three architectures that are used in the rest of the paper. However, after reading the section, it is not clear what the essential components are. Given that the introduction of the task is a core contribution of the paper, a more structured description would be really appreciated, especially the details concerning continual learning. This should also go in the main section of the paper (it could be short, but it should be complete). Also as a good example of what I mean is Schuessler et al. (2020), which opens section 2 with a very concise description of the task.

	**Minor presentation issues**
	At a different level, the paper would benefit from:
	- less strange wording in many cases (e.g “more powerful performance-optimized continuously nonlinear RNNs”),
	- other small fixes (numbering equations, normalizing the references (use \citep in latex), fix a number of typos (e.g. “gradient decent” last paragraph in Section 1), etc).

**References**

Chaudhry, Arslan, Marc’Aurelio Ranzato, Marcus Rohrbach and Mohamed Elhoseiny (2019) Efficient Lifelong Learning with A-GEM. *ICLR 2019*

Schuessler, Friedrich, Francesca Mastrogiuseppe, Alexis Dubreuil, Srdjan Ostojic, and Omri Barak (2020) The interplay between randomness and structure during learning in RNNs. *NeurIPS 2020*

**Summary Of The Paper:**

This paper considers the problem of RNNs learning arbitrary sequences of “motifs” (continuous time functions, discretized) after having learned the individual motifs separately. Inspired by some previous work, it proposes an architectural approach to deal with interference by introducing a motif-dependent low-rank perturbation to the recurrent weights of the RNN, following Schuessler et al. (2020). To this end, the paper introduces a new evaluation task (motif generation), which is used to compare that architecture and sensible baselines. It then moves to propose a new model inspired in thalamocortical interactions in the brain: an RNN equipped with a preparatory module that imposes a beneficial bias when following a specific training protocol.

**Summary Of The Review:**

The general topic of this paper is relevant, but I found the presentation confusing and difficult to understand. Overall, given the current state of the presentation, some issues with the details of the continual learning experiments, and some concerns related to the existence of a trivial solution that would need to be addressed, the contributions of this paper remains unclear to me and I suggest a rejection.

---

> ### Author Response · Authors · 2021-11-16
> **clarifications: part 1/2**
>
> We thank the reviewer for their review, and for their constructive comments to improve the presentation of our results. We first clarify some technical points below.
>
> # Technical
>
> * First, concerning continual learning, we work with a qualitatively different type of solution than Chaudhry et al., ICLR, 2019 - we will explain why the metrics used in this paper are not applicable in our case, and *which metrics we used*. The approaches exemplified by Chaudhry et al. attempt to find gradient updates that minimize interference among tasks. In this case, measures that quantify the amount of forgetting (as in Chaudhry et al.) are meaningful. In contrast, we focus on a modular architectural approach to continual learning. This involves building *(i)* modules whose parameters are segregated across tasks (e.g. Parisi et al., 2019; Rusu et al. 2016; Hadsell et al, 2020) and *(ii)* at least one module whose parameters are not updated when learning new tasks (but that can be pre-tuned, e.g. to make sure it solves the first task) so that it can be re-used across tasks. In this approach, *interference is always fully prevented by construction* as each task is learned by gradient descent updates that exclusively act on the parameters of the specific module it is assigned to, such that gradient descent updates never overlap across tasks. The evaluation of this type of approach relies on measuring whether the network makes efficient use of the parameters: how much does segregating the task-tuned parameters per module impact the *capacity* of the network?
>
>   To evaluate this, we were able to rely on the existing literature that has studied the capacity of different RNN architectures in a general setting (Collins et al. 2017), even though - to the best of our knowledge - such a modular approach to RNN design for continual learning had not been tried before. Indeed, Collins et al. found that the capacity of non-modular RNNs scales as their number of tuned parameters. To test whether introducing modules to segregate parameters per motif decreased capacity in our setting, we compared the performances of our parameter-segregated architectures (additive and multiplicative) to a control architecture - whose parameter count is adjusted to equalize the number of tuned parameters per motif across architectures, but which does not segregate the parameters per motif. The control architecture does not have any mechanism that would protect against learning interference - accordingly, as we mentioned in section 1, we found that when learning motifs sequentially with this network, previously learned motifs are immediately forgotten when learning a new one - while the parameter-segregated architectures do not forget at all. We did not show this in a figure because it is expected, but we'll be happy to include this in a revision. Importantly, however, we did quantify the accuracy of these different architectures matched in number of tuned parameters when including all possible two-motif sequences in each training batch (Fig. 1c). By using this metric, we find that segregating motif-tuned parameters has little impact on network capacity, validating our parameter-segregated architectures as efficient solutions to continual learning in our task.

---

> > ### Author Response · Authors · 2021-11-16
> > **clarifications: part 2/2**
> >
> > * In addition, we regret that we did not manage to explain that our results clearly support all the three points in the last paragraph of section 3, explaining the drawbacks of a hard reset strategy.
> >
> >   First, we note that hard arbitrary resets prevent the production of naturally interpolating motor sequences - which strongly impact the performance of the motor effector down the line (think how hard it would be to speak if we had to reset our face and larynx muscles before each new syllable). Instead, our preparatory module supports the production of naturally interpolating transitions even if the end and start points of motifs vary (as is more notably visible in Fig. 14, panels 'When starting from other motifs' and 'Transitions from $70$% of the prev. motif').  This is because our preparatory module *does not drive the RNN to the origin when it is combined with a motif-specific input to implement motif preparation* (e.g. see the varied output values in Fig. 3d at the end of the preparatory periods outlined by grey boxes). Instead, in our networks with a preparatory period, the internal state directly interpolates between an arbitrary activity pattern at the end of the previous motif and a motif-specific pattern at the beginning of the next motif, which supports naturally interpolating transitions.
> >
> >   Second, even if setting the question of smooth interpolation at transitions aside, our results speak to other limitations of simply starting motifs with a reset of the network state, which we can study by training and testing the network to produce single motifs when starting from a fixed type of initialization. Notably, our random Gaussian initialization exactly corresponds to a noisy reset to the zero state (which is suitable for our motifs that start at 0), and in this scenario we show that the preparatory module helps the networks to learn more accurate motifs (Fig. 3c left; see the third point of our response to reviewer MvgU for details).
> > Relative to the possibility to reset to a noiseless zero-state (point *(i)* in the last paragraph of section 3 mentioned by the reviewer), we actually have extensive experience showing that in general, adding noise during training is necessary to prevent extreme noise sensitivity in RNNs - this was so natural to us that we did not think it would be interesting to describe it in detail. For instance, after training an additive network to produce single motifs from a noiseless zero state initialization, not only did this result in a very large noise-sensitivity, but it also required about twice as many trials to reach plateau performance compared to the additive network with a preparatory module 'handicapped' by requiring robustness to random initialization during training (and the plateau performances were similar in the two cases, even appearing slightly worse for the RNN trained from a noiseless zero state). We will be happy to add these details to the revised manuscript, which emphasize again the combined advantages of the preparatory module to efficiently learn accurate motifs (illustrated in Fig. 3c left) and to improve noise robustness and generalization (illustrated in Fig. 3c right).
> >
> > &nbsp;
> >
> > # Writing:
> >
> > We thank the reviewer for their useful remarks. Evidently, our attempts to simply define the task in the introduction and in Fig. 1a were unsuccessful. We also recognize that the definition that we included in the appendix - whose wording and structure were developed with a computer science researcher in an attempt to formally lay down the constraints of our task - is not well integrated with the rest of the text. We will work on incorporating the reviewer's ideas in our revised manuscript.
> >
> > &nbsp;
> >
> > **References**
> >
> > German I. Parisi, Ronald Kemker, Jose L. Part, Christopher Kanan, and Stefan Wermter. Continual lifelong learning with neural networks: A review. Neural Networks, May 2019.
> >
> > Andrei A. Rusu and Neil C. Rabinowitz and Guillaume Desjardins and Hubert Soyer and James Kirkpatrick and Koray Kavukcuoglu and Razvan Pascanu and Raia Hadsell, Progressive Neural Networks, arXiv, 2016.
> >
> > Raia Hadsell, Dushyant Rao, Andrei A. Rusu, Razvan Pascanu, Embracing Change: Continual Learning in Deep Neural Networks, Trends in Cognitive Sciences, Volume 24, Issue 12, 2020.
> >
> > Jasmine Collins, Jascha Sohl-Dickstein, and David Sussillo. Capacity and trainability in recurrent neural networks. In 5th International Conference on Learning Representations, ICLR 2017.

---

### Official Review · Reviewer_hBJi · 2021-11-02

**Correctness:** 4
**Technical Novelty And Significance:** 3
**Empirical Novelty And Significance:** 4
**Recommendation:** 6
**Confidence:** 2

**Main Review:**

Pros
- The modification that's required to the "control" network in order is extremely simple, and has a clear theoretical goal of decreasing "residual" activity between motif transitions, without quenching activity in response to the new control signal.
- The level of experimental detail is sufficient such that the experiments  in the main text can be replicated by an independent reader.

Cons
- The methods in the main text are clear, and lead to convincing results in the main-section. However, without digging much deeper into appendix 4 & 5, there's no clear reason why the learned preparatory modules are able function

**Summary Of The Paper:**

The article proposes a biologically inspired architecture which increases the degree to which simple recurrent neural networks may be able to chain together elemental output sequences. The authors show that the proposed "predatory module" allows networks to perform on par as if the network were starting from a set initial state, while also providing a smooth transition between motifs.

**Summary Of The Review:**

The preparatory module appears to function well, and the quenching activity is reasonable well tied to biological thalamocortical loop actions. To my knowledge, the motif-transition problem is not solved in switching recurrent networks, however I am not overly familiar with that literature. Overall, my low confidence ratings come from not knowing how interesting or novel the analytical work in the appendices is.

---

> ### Author Response · Authors · 2021-11-16
> **clarification about the novelty and relevance of the work**
>
> We thank the reviewer for their comments. In a revision, we will further clarify our approach, and better explain the main novelty and interest of our work: showing that theoretical insights derived in a constrained biological model can be useful during gradient-descent training of RNNs of the type used by some members of the machine learning community - which are notably free to hit a nonlinearity at any time, while the original model isn't. This is not trivial, for two reasons. First, these 'fully' nonlinear RNNs operate beyond the dynamical regime for which the theory is valid - which notably increases their expressivity and presents concrete advantages in our task such as very accurate transitions. Second, the theory did not cover how an online learning rule could be used to learn the weights of the model. Therefore, our finding that infusing 'fully' nonlinear RNNs with biological insights leads to very efficient gradient-descent-based training was far from guaranteed and is key for any future application of the framework.
>
> We were surprised and pleased to discover that our trained biologically-inspired RNNs get the best of both worlds: the larger expressivity and versatility afforded by the richer dynamics of the 'fully' nonlinear RNNs trained with gradient descent, and the robustness and generalization capacities of the simpler analytically tractable model that was constrained by biology. This is exciting for us because, in our experience, there are very few insights from biologically-constrained models that translate into improved performance for gradient-trained networks that are not constrained to operate in a limited dynamical regime. The interest of these results is not restricted to the design of the networks that can efficiently and robustly learn to generate flexible sequences: they reveal that the widespread biological phenomenon of motor preparation may reflect a general computational operation rather than result from dynamics constraints that arose through evolutionary idiosyncracies.

---

### Official Review · Reviewer_MvgU · 2021-11-03

**Correctness:** 3
**Technical Novelty And Significance:** 2
**Empirical Novelty And Significance:** 2
**Recommendation:** 3
**Confidence:** 4

**Main Review:**

The authors discuss two important alternatives to the approach that they ultimate pursue, which would involve attacking the problem with learning rather than with building in inductive biases. Rather than running experiments to test the efficacy of these alternatives, the authors instead tried to argue away these points (bottom of pg. 6). This is extremely unsatisfying.

Learning has proven remarkably powerful, and incorporating constraints/structure/inductive biases, may indeed be necessary in some domains to overcome current challenges, as the work the authors cite suggests. But those domains are ones in which learning has been doggedly pursued for an extended period of time. These authors instead introduce a novel synthetic task and immediately jump to the need to incorporate more structure, without doing the legwork necessary to convincingly demonstrate that such structure is indeed necessary.

To directly respond to the authors arguments over why these alternatives are insufficient:
First, they suggest that directly training the RNNs to generalize from end states is insufficient, because they can't train from all possible end states, particularly as the number of trained states scales. This argument is unpersuasive, as training on all of anything is often impossible in general, yet still learned systems generalize. E.g., work with ImageNet over the last decade provides a great example of this.

The authors could instead test this possibility by incorporating the end points into training as initial states for subsequent trajectories. They could hold out some transitions and test how well the system generates the motifs on these left-out transitions. They could hold out the transitions in two ways: Either exclude some transitions randomly (i.e., salt and pepper of the transition matrix) or hold out some motifs from the training at all (i.e., holding out rows or columns of the motifs). The latter would be a stronger test, but both would be important to see. Critically, for the authors to convincingly show that learning was insufficient, they would need far, far more than 10 motifs (e.g., like >>1000, most likely).

Second, the authors could train to return to a consistent point at the end of each motif. The authors assert that doing so would make the system not robust to perturbation/noise, but that would be true only if the system actually learned to arrive at an identical point each time, which I imagine would not occur (and if they did, they could add noise to the network during training). The network could return to the origin imperfectly and convey state information (e.g., on 299 dimensions return to origin, but on one axis keep information about). Who knows if these speculations are correct -- but an experimental test of these ideas would be quite useful.

My second major concern is the relevance of this work to the broader field. The networks are small and seemingly toy: 300 units, 10 motifs. This is smaller than MNIST. I'm not one to think that everything needs to be shown on ImageNet and achieve SOTA to be a useful contribution, but the scale of these experiments are so, so small that I worry that anything learned in this domain would transfer or inform work at far greater scales -- particularly when the conclusion is that in such a low data regime, greater inductive biases are required rather than learning.


Minor issues:

- The authors assert that thalamic activity may change the effective connectivity of motor cortical activity, but they don't make clear why that change would result in a bias towards the origin. In what sense is biasing the network to the origin motivated by neuroscience? Also, how is preparatory activity being the same across all motifs brain-like? Isn't that not seen in the brain (e.g., in Susillo et al. 2015)?

- The authors seem to repeatedly suggest that matching the marginals of a distribution is a strong approximation to the joint -- e.g., "Subtle variations in the state of the network at the end of different first motifs x_{end}^{\mu} can lead to very different outcomes". I don't understand why the authors are overstating the extent to which the marginals and the joint are the same thing...

- The authors assert that "a standard normal is a good choice" for the marginals of the states. Please provide some quantification for this claim. As it stands, the authors visualize cumulative distributions of a subset of the states rather than performing statistics.

- How did the authors pick a standard deviation of 1.2 (in Fig 2.a.iv-v)? Did they fit this number to the marginals?

- Typo, top of page 5: "require less neurons" --> "require fewer neurons"

**Summary Of The Paper:**

The authors seek to use a small RNN to generate extended, time-varying outputs given a cue indicating which output to generate. They focus on the problem of generating a sequence of outputs, cued in series, without the end state of one interfering with the subsequently cued trajectory. They show that small vanilla RNNs do not readily perform this task: RNNs at a random state can generate a sequence when cued, but they fail to do so when their state is at the end point of another sequence. The authors then show that by biasing the activity of the network towards the origin in the absence of an input (which they suggest is inspired by neuroscience), RNNs more readily learn sequences and more consistently generate accurate sequences in series. My enthusiasm for the work is relatively low for a variety of reasons. Primarily, the authors fail to test obvious alternative hypotheses of ways that the problem domain they design could be solved by a learned system, without the need to add the inductive bias they add. Moreover, and more fundamentally, the relevance of such a small toy system (300 units, trained to generate just 10 trajectories) for machine learning is unclear.


**Summary Of The Review:**

The small scale of the models/data combined with the lack of evaluation of alternative approaches diminishes the potential contribution of this work.

---

> ### Author Response · Authors · 2021-11-16
> **clarifications: part 1/2**
>
> We thank the reviewer for their comments, which show that we failed to explain some of our key results and how they relate to the constraints of our motor sequencing task. We address their concerns below, and will make sure to include the related clarifications in a revision of the manuscript.
>
> &nbsp;
>
> * Importantly, we *verified that increasing the number of units does not improve the ability of the RNNs to transfer from single motif training to sequencing* (RNN of 1000 units shown in Fig. 2c) -  besides, our baseline number of units is actually larger than in previous related ML works (e.g. Schuessler et al., Maheswaranathan et al.). In addition, for each of our data set, the relatively small number of motifs we used did not prevent us from thoroughly evaluating the performance of our networks because our solution involves learning different motifs fully independently, and therefore the performance is statistically equivalent for the first, the tenth, or the 10 0000th learned motif (more on that below).
>
> &nbsp;
>
> * We regret that we failed to communicate why our problem constraints and our results imply that a network that can generalize from single motif training to performing motif sequences is much preferable to a network that needs motif transition training to function. Indeed, for motor control in the real world, the ability to improvise new sequence orders zero-shot is often key to avoid negative consequences - for instance, the ability of a bouldering climber to use a proper landing movement regardless of which movement they were performing before falling often prevents injury.
>
>   The reviewer suggests adding some transitions to the training set and testing on held-out transitions. We recognize that in the text we only discussed this question in light of our observation that variations in end-of-motif activity patterns can lead to vastly different outcomes for the next motif (from quasi-flawless to very inaccurate, see Fig. 9), so that efficiently sampling all of these varied patterns during training appears difficult. Beyond this, we want to clarify here that *including a subset of sequences during training also disrupts the sequential training strategy* that prevents the training set to grow over time in a continual learning setting. Indeed, on the one hand, sequential training would require motif parameters to only be trained for transitioning from previously learned motifs - which would compromise the robustness of the motifs that were learned earlier, especially if the motifs' properties change over the 'life' of the agent. On the other hand, re-training earlier-learned motifs to improve their accuracy when they are initialized from later-learned motifs causes the training set to grow with the number of learned motifs, and runs the risk of modifying the network's end-of-motif activity for these re-trained motifs, which could in turn affect the transitioning from these motifs to any others.
>
>   We will be happy to clarify this in a revision, but the above indicates that *sequence learning strategy introduces drawbacks and risks* that are not present when using our preparatory module to train single motifs. *This does not take anything away from the generalization abilities of learning-based strategies - notably, here, our RNNs with a preparatory module can only perform well during sequencing because of the exceptional generalization abilities of gradient descent training when deployed on an appropriate architecture* (in a similar way that transformers generalize well thanks to both architectural constraints and gradient descent training). Therefore, we report a non-trivial and useful network design for continually learned motor motifs by demonstrating a way to transfer zero-shot from single motif training (e.g. landing after a fall from a safe position on a climbing wall) to motif sequencing (e.g. first unplanned fall from an arbitrary climbing move).

---

> > ### Author Response · Authors · 2021-11-16
> > **clarifications: part 2/2**
> >
> > * The reviewer mentions a possible strategy where the network activity would be constrained during learning to end at the same point (with noise) for all motifs - so that all motifs would be started from a consistent state even during sequencing.
> >
> >   It seems that we have to clarify that any candidate alternative strategy would have to match the ability of our approach to naturally interpolate between variable end-of-motif values and variable start-of-motif values (see notably Fig. 14, panels 'When starting from other motifs' and 'Transitions from $70$% of the prev. motif').  This is because our preparatory module *does not drive the RNN to the origin when it is combined with a motif-specific input to implement motif preparation* (e.g. see the varied output values in Fig. 3d at the end of the preparatory periods outlined by grey boxes). Instead, in our networks with a preparatory module, the internal state directly interpolates between an arbitrary activity pattern at the end of the previous motif and a motif-specific pattern at the beginning of the next motif - without the need for a detour by a motif-independent state, which would be inefficient and potentially dangerous for the motor effector which reads out from the network. Given that the motor effector must readout from the modulable dynamics of an RNN at all times, we do not see how the issue of smooth and accurate transitions could be solved zero-shot using the reviewer's idea of forcing most of the RNN's activities to end at the same point for all motifs (even if creating separate variables to remember the network state before transitions).
> >
> >   In addition, even in the context of starting motifs from a noisy consistent state, our results demonstrate a separate important advantage of our preparatory module. Indeed, this is essentially what we investigated when we both trained and tested our networks on isolated motifs with a random Gaussian initialization. We showed that the baseline networks (without a preparatory module) had a much larger error than the networks with a preparatory module (Fig. 3c left, full vs. dashed lines). An examination of the networks' readout shows that, beyond an increased difficulty to filter out noise, a major driver of this increased error is an inaccurate average motif shape (Fig. 3d, bottom two rows). This is strong evidence that our preparatory module helps in two ways in this scenario. First, when training single motifs from a fixed noisy initialization, the fast dynamics timescales introduced by preparatory module help the training-induced changes in the input to drive the exploration of different possible activity states until finding a motif-specific one that leads to particularly accurate motif production (Mohajerin and Waslander, 2019; Kemeth et al., 2021). Second, it helps filtering out noise and robustly converging to this motif-specific activity pattern to start motif production.
> >
> > &nbsp;
> >
> > * From various comments of the reviewer, it appears that we may have failed to stress that the statistics of the end-of-motif network activity distributions are shaped during learning - they cannot be determined from a-priori accessible data, they can change in response to changing the initial activity pattern used during training, and they are motif-specific. This is why the problem we tackle is non-trivial. The reason why we emphasize the role of correlations in the activity pattern is that these are often neglected in mean-field theoretical approaches to understanding RNN dynamics.

---

> > > ### Comment · Reviewer_MvgU · 2021-11-29
> > > **Response to comments**
> > >
> > > I have reviewed the authors' comments and looked over the updated manuscript. I appreciate their efforts and they have addressed parts of my concerns, but I still stand by my original scores.

---

### Official Review · Reviewer_BooE · 2021-11-03

**Correctness:** 3
**Technical Novelty And Significance:** 2
**Empirical Novelty And Significance:** 2
**Recommendation:** 5
**Confidence:** 3

**Main Review:**

Strength: The proposed architectures have been shown to work for "zero-shot learning," or recombination without re-learning.

Weakness: It is not convincing whether the particular resetting mechanism is the most reasonable one, either computationally or biologically. Computationally, any strong inhibitory input to bring the network state to the origin should suffice. Biologically, the most likely mechanism is the tonic inhibition and timely disinhibition by the basal ganglia to the thalamus and subcortical motor controllers like the superior colliculus.

**Summary Of The Paper:**

This paper presents the problems in chaining nonlinear motor pattern generators and proposes a resetting mechanism inspired by the thalamo-cortical network.

**Summary Of The Review:**

An interesting proposal worth discussion.

---

> ### Author Response · Authors · 2021-11-16
> **clarifications**
>
> We thank the reviewer for their comments, and we clarify below how our results address the reviewers' concerns.
>
> &nbsp;
>
> * In our $tanh$ networks, a strong inhibitory input would not drive our networks to the origin.
>
>   Regardless, any mechanism that would implement such a naive 'reset' at motif transitions to any fixed state like the origin (after using this state to initialize networks during single motif training) would have substantial drawbacks compared to using our preparatory module.
>
>   Indeed, our preparatory module - *when it is combined with a motif-specific input to implement motif preparation* - actually leads the networks towards an appropriate *motif-specific* state (e.g. see the varied output values in Fig. 3d at the end of the preparatory periods outlined by grey boxes).
> Therefore, in contrast to a reset strategy, our solution leads to accurate and smooth transitions between motifs of arbitrary start and end points as it does not make a detour through an arbitrary state like the origin (e.g. see Fig. 14 panels 'When starting from other motifs' and 'Transitions from $70$% of the prev. motif').
>
>   Besides, the 'naive reset strategy' - that forgoes the preparatory module - also leads to impaired single motif learning. Indeed, on the one hand, we have observed that training motifs from a fixed noiseless state leads to extreme noise sensitivity (we will show this in a revision; see the second part of our technical response to reviewer W1np for further details). On the other hand, we have shown in the article the result of naively training from a fixed state (the origin) while adding noise: this exactly corresponds to the performance of the additive and multiplicative networks with random Gaussian initialization (Fig. 3c left, full lines). The results show that, if forgoing the fast adjustment of the network state towards an appropriate motif-specific pattern enabled by the preparatory module, networks are much less accurate (Fig. 3c left, full vs. dashed lines; see the third part of our response to reviewer MvgU for further details).
>
> &nbsp;
>
> * The reviewer states that the preparatory module is not biologically plausible. Previous works we cite (Kao et al 2021; Logiaco et al 2021) have demonstrated the link between the characteristics of our preparatory module and the brain's architecture and dynamics. We included the related details in the appendix: the preparatory module implements thalamocortical loops. The involvement of strong thalamocortical interactions specifically during the preparatory period of precise arm movements has indeed been recently demonstrated experimentally (Nashef et al 2021), which validates the biological plausibility of our circuit for our fine autonomous motor control task. More generally, the superior colliculus (mentioned by the reviewer) tends to be involved in relatively simple, quasi-automatic sensorimotor transformations, while a large body of evidence supports the involvement of motor cortex for more complex learned movements (e.g. Sauerbrei et al., Cortical pattern generation during dexterous movement is input-driven, Nature 2019; and the monkey lesion literature).

---

### Author Response · Authors · 2021-11-24
**updated manuscript**

To address the reviewers' comments, we have uploaded an improved draft that both (i) implements necessary clarifications; and (ii) presents more evidence confirming the advantages of using our preparatory module for motif transitions relative to alternatives:

* We clarify, and provide more evidence for, the advantage of using our preparatory module compared to using any reset strategy to a fixed state (related to the first point of our response to Rev. BooE, the third point of our response to Rev. MvgU, and the second point of our technical response to Rev. W1np). These modifications are on the top of page 7 and in section 4.

  First, we clarify that a reset to a fixed state forces the motor effector to make costly detours when transitioning between motifs of varied start and endpoints (top of page 7), while our preparatory module enables agile transitions that directly interpolate between these points (paragraph 'Second' on page 8, now referring to a clarified Fig. 14 legend that directly shows this).

  Second, we provide further evidence that even when training motifs from a 'reset state', our preparatory module enhances gradient-descent training of motif-specific parameters for improved accuracy and robustness during motif production. We describe at the top of page 7 how a reset to a noiseless state would not be noise-robust, and refer to a new Fig. 16 (in section A.6.3) that directly shows this.  We also explicitly explain how our results on training and testing from randomly initialized RNNs correspond to a noisy reset strategy (top of page 7) and how - even in these conditions - our preparatory module synergizes with gradient-descent training of motif-specific parameters not only to improve the accuracy and robustness of motif production, but also for speeding up learning (paragraph 'First' on page 8). For this latter point specifically, we refer to a new Fig. 17 and associated section A.6.4.

* We clarified how the continual learning constraints of our task are discordant with introducing motif sequences in the training set (last paragraph of page 6, related to the second and last points of our response to Rev. MvgU)

* We clarified which metrics we used to validate our architectural approach to continual learning and why the RNNs we propose do not suffer from any interference during our single motifs training strategy (top text on page 3 and last paragraph of section 2; related to the first point of our technical response to Rev. W1np).

* We implemented various recommendations of Rev. W1np to improve the presentation of our results (in section 2, Appendix A.6.1 and Appendix A1).

We thank the reviewers for their feedback. We feel that by addressing the reviewers' concerns as described above, we really improved the manuscript and demonstrated the relevance and validity of our work. We are looking forward to any future discussions.

---

### Decision · Program_Chairs · 2022-01-20

**Decision:**

Reject

**Comment:**

This manuscript presents a method to allow RNNs to chain together sequences of behaviors. Reviewers had numerous concerns but the most important is that the problem posed here is solved by a simple method: resetting the state of the RNN before processing a motif.

Overall, reviewers noted a few key topics, although this list is not exhaustive:
1. Experiments are in a very simple but confusing setting.
2. Even though alternatives exist to solving this problem, they are not considered.
3. The networks considered are very simple.
4. The manuscript is difficult to understand.
5. The task admits a trivial solution.

In more detail:

1. The setting of learning to memorize time series and outputting them on command is very simple compared to what most modern work considers. Moreover, there is much confusion in the manuscript about what the setting is precisely.  For example, the setting is described as "independently learn motor motifs in order to build a continuously expandable motif library". But there is no continuously expandable motif library, the motif library is fixed at test time. The authors focus heavily on calling this setting "motor motifs", but these are RNNs that output an arbitrary time series. They are in no sense motor programs and this work is not connected to the extensive literature on motor control in machine learning. More broadly, there is no clear mathematical definition of what the problem being solved is anywhere in the manuscript.

2. It is unusual for manuscripts to not present other baseline models. But more importantly, many other approaches exist to this problem. As one reviewer pointed out, the manuscript essentially sets out to solve a problem that is completely solved in machine learning today. It rejects the solutions that exist for arbitrary reasons, and then adopts its own new solution.

3. The models used are very simple, but this is a consequence of 1, the problem domain being very simple.

4. Reviewers had difficulty understanding the details of the task. In particular, the task description section begins with minutia about the implementation rather than succinct definition of the task.

5. Most critically, reviewers identified that the model could be hard reset and would have the same behavior as the model presented in the manuscript. The proposed solution is essentially hard resetting the state to zero as it stands. There is no reason why a hard reset cannot be followed by a smoothing operation -- this seems to be the main objection of the authors.

Overall, the manuscript needs significant improvements. The task considered is too simple by modern ML standards and the fact that it admits a simple solution cannot be overlooked. Demonstrating the idea of the preparatory module on an existing ML task and dataset, while comparing with existing baseline models, carrying out ablations, and producing an extensive quantitative evaluation is what will get the community excited about preparatory modules.

---

> ### Public Comment · ~Laureline_Logiaco1 · 2022-11-21
> **misunderstandings on the nature of our task, and on how it can and cannot be solved**
>
> We regret that we were not able to explain the nature of the problem that we solve, and why this specific problem (i) has - to the best of our knowledge - neither been addressed nor solved by the machine learning literature focusing on motor control (despite our extensive literature review, as we clarify in the first paragraph below); and (ii) cannot be circumvented by hard resets (as we clarify in the second paragraph below).
>
> The difficulty of our problem lies in the improvisation of new sequence orders in networks trained to produce individual continuous motifs; specifically, autonomously produced ones that animals can perform when sensory feedback is too slow and/or unreliable (see e.g. Fig. 4 in Rothwell JC, Traub MM, Day BL, Obeso JA, Thomas PK, Marsden CD. Manual motor performance in a deafferented man. Brain. 1982 Sep;105 (Pt 3):515-42. doi: 10.1093/brain/105.3.515. PMID: 6286035). Therefore, in contrast to the focus of the many papers from the machine learning literature we reviewed and referenced, our work does not aim at pushing the boundaries of the transformation from complex sensory input to a 'reactive' motor output for either a fixed motor program and/or rehearsed sequences of motifs. In consequence, we do not believe that existing benchmarks are appropriate to assess our networks. Importantly, we did test our networks by performing 'ablation' experiments to identify the important features of our proposed solution. In addition, we did vary the complexity of our autonomous motor motifs and checked that our results are stronger for more complex motifs.
>
> Specifically, we were able to build networks that can extendably and sequentially learn a library of autonomously produced continuous motifs, and can - with no additional training - improvise smooth and accurate motif transitions to flexibly recombine these motifs into motor chains. A hard reset of the network state at transitions combined with a causal filter of the network output (to try to smooth out discontinuities between the end of the previous motif and the start of the new motif) would not solve this task in a satisfying manner, as it would notably introduce highly undesirable distortions and limitations in the generated output (among other issues - discussed in detail in our response to reviewers - that disqualify hard resets in our context). Our proposed mechanism circumvents these issues precisely because it is not implementing such a reset of the network to zero (or to any pre-determined network state). This is a good illustration of how some methods such as hard resets, which are designed for input-driven and/or mostly feedforward networks, do not apply to networks solving our biologically-motivated task of autonomously and flexibly producing continuous outputs. Such an ability to improvise motif sequences, while producing timely and accurate motor output, can be life-saving for animals in the real world - for instance for successfully executing a landing move after falling at the end of a long sequence of climbing movements, even though this landing move had only been practiced from a safe position.